# d3LLM: Ultra-Fast Diffusion LLM using Pseudo-Trajectory Distillation

**Yu-Yang Qian**[1,2] **Junda Su**[1] **Lanxiang Hu**[1] **Peiyuan Zhang**[1] **Zhijie Deng**[4] **Peng Zhao**[†,2,3] **Hao Zhang**[†,1]

## Abstract

Diffusion large language models (dLLMs) offer capabilities beyond those of autoregressive (AR) LLMs, such as parallel decoding and random-order generation. However, realizing these benefits in practice is non-trivial, as dLLMs inherently face an *accuracy-parallelism trade-off*. Despite increasing interest, existing methods typically focus on only one-side of the coin, targeting either efficiency or accuracy. To address this limitation, we propose d3LLM (*Pseudo-Distilled Diffusion Large Language Model*), striking a balance between accuracy and parallelism: (i) during training, we introduce *pseudo-trajectory distillation* to teach the model which tokens can be decoded confidently at early steps, thereby improving parallelism; (ii) during inference, we employ *entropy-based multi-block decoding* with a KV-cache refresh mechanism to achieve high parallelism while maintaining accuracy. To better evaluate dLLMs, we also introduce AUP (*Accuracy Under Parallelism*), a new metric that jointly measures accuracy and parallelism. Experiments demonstrate that our d3LLM achieves up to $10\times$ speedup over vanilla LLaDA/Dream, and $5\times$ speedup over AR models without much accuracy drop. Our code is available at https://github.com/hao-ai-lab/d3LLM.

## 1. Introduction

Diffusion large language models (dLLMs) have emerged as a promising alternative to autoregressive (AR) LLMs. A key advantage of dLLMs is their use of *bidirectional attention*, which enables capabilities such as parallel decoding, error correction, and random-order generation—features that are not feasible with AR models. Recently, several closed-source diffusion models, including Mercury (Khanna et al., 2025), Gemini Diffusion (Google DeepMind, 2025), and Seed Diffusion (Song et al., 2025), have demonstrated impressive efficiency and performance, achieving extremely high throughput and sometimes exceeding 1000 tokens per second in certain settings. In contrast, open-source dLLMs have exhibited significantly lower throughput, sometimes even slower than AR baselines. For example, LLaDA (Nie et al., 2025) and Dream (Ye et al., 2025) achieve only around 20 tokens per second. Moreover, they often lag behind similarly-sized AR models in terms of accuracy.

With growing interest from the research community, an increasing number of methods have been proposed to accelerate dLLMs (Wu et al., 2025b; Wang et al., 2025a; Ma et al., 2025a; Chen et al., 2025; Wu et al., 2025a; Ma et al., 2025b). The most state-of-the-art algorithms are Fast-dLLM-v2 (Wu et al., 2025a), which converts AR models into dLLMs by fine-tuning them with a block diffusion mechanism and complementary attention mask, and dParallel (Chen et al., 2025), which introduces a certainty-forcing distillation algorithm to enable the dLLM to decode more tokens at a time. They achieve nearly twice the throughput compared with AR baselines. Another line of work focuses on improving the performance of dLLMs (Yang et al., 2025b; Bie et al., 2025; Cheng et al., 2025), typically by employing more advanced training strategies, extending context length and multimodal capabilities, incorporating reasoning abilities, and collecting larger or higher-quality datasets.

However, previous works *typically focus on only one-side of the coin*, targeting either efficiency or performance. For example, in our empirical evaluation in Section 4, methods such as D2F (Wang et al., 2025a) prioritize parallelism but incur a notable accuracy loss compared to similarly sized AR models, whereas methods like Fast-dLLM-v2 (Wu et al., 2025a) preserve accuracy at the cost of reduced parallelism. In other words, most improvements to dLLMs implicitly slide along a trade-off frontier: greater parallelism typically causes lower accuracy, and vice versa. We argue that this trade-off is fundamental, as dLLMs *by nature live on an accuracy–parallelism curve*.

Consequently, this observation raises a natural question: *how can we push the accuracy–parallelism frontier further?*

[1]University of California, San Diego [2]School of Artificial Intelligence, Nanjing University [3]State Key Laboratory for Novel Software Technology, Nanjing University [4]Shanghai Jiao Tong University. Correspondence to: Peng Zhao <zhaop@lamda.nju.edu.cn>, Hao Zhang <haozhang@ucsd.edu>.

*Proceedings of the 43rd International Conference on Machine Learning*, Seoul, South Korea. PMLR 306, 2026. Copyright 2026 by the author(s).

To answer this, we first identify two limitations of existing dLLMs that underlie this trade-off: (i) regarding *training*, standard dLLM training employs random masking, which provides no guidance on which tokens can be safely decoded earlier with high confidence. As a result, when attempting to decode more tokens in parallel, the model inevitably unmasks uncertain tokens prematurely, degrading accuracy; (ii) regarding *inference*, traditional decoding methods focus on generation within a single block, inherently limiting parallelism. While recent work (Wang et al., 2025a) extends this to multi-block decoding to improve parallelism, it tends to degrade generation quality because predictions in later blocks rely on incomplete or erroneous context from preceding blocks. In short, existing methods face a dilemma: pushing for higher parallelism compromises accuracy; while preserving accuracy constrains parallelism.

To address these limitations, we propose d3LLM (*pseuDo-Distilled-Diffusion LLM*), striking a balance between accuracy and parallelism. (i) At training time, we introduce *pseudo-trajectory distillation*: rather than relying solely on ground-truth outputs, we incorporate the teacher dLLM's own decoding trajectory. This provides intermediate supervision, indicating which tokens can be safely decoded earlier. We further incorporate a *curriculum learning strategy* that progressively increases the difficulty. (ii) At inference time, we introduce *entropy-based multi-block decoding* that simultaneously decodes across multiple blocks by prioritizing low-entropy (high-confidence) tokens. To mitigate quality degradation, we employ a *KV-cache refresh mechanism* that periodically recomputes all previously cached states.

We validate the effectiveness of d3LLM on three representative open-source foundation dLLMs: LLaDA (Nie et al., 2025), Dream (Ye et al., 2025), and Dream-Coder (Xie et al., 2025), using our proposed new metric AUP (*Accuracy Under Parallelism*), which jointly captures both generation quality and parallelism. Experimental results show that d3LLM achieves the highest AUP score on 9 out of 10 tasks, delivers $3.6\times$–$5\times$ speedup over AR models (Qwen-2.5-7B-it) depending on the GPU platform, and attains $10\times$ speedup compared to vanilla LLaDA/Dream, with negligible accuracy degradation. Moreover, for the more challenging *coding generation* scenario, we are the first to develop an efficient dLLM-coder with performance comparable to AR coders, achieving $8\times$ speedup over vanilla Dream-Coder.

**Organization.** Section 2 introduces problem formulation. Section 3 presents our d3LLM framework. Section 4 reports experimental results, and Section 5 concludes the paper.

## 2. Problem Formulation

In this section, we introduce our evaluation metric for dLLMs. Our observation is that the literature tends to re-

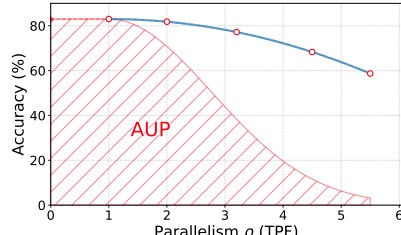

**Figure 1.** Illustration of the AUP metric, where we calculate the weighted area under the accuracy-parallelism curve.

port diffusion progress using single, isolated metrics, such as efficiency-only metrics like *tokens per second* (TPS) / *tokens per forward* (TPF), or performance-only metrics like accuracy (solve rate / pass@1 accuracy). However, a key insight is that, unlike AR models, dLLMs naturally *live on an accuracy–parallelism curve*. Consequently, single metrics become misleading, as they overlook the fundamental trade-off between efficiency and performance and fail to answer the real question: How well does a method maintain accuracy as we push parallelism higher? These insights motivate us to design a new unified metric.

In fact, most dLLM methods already expose certain knobs that trade off speed and quality. For example, FastdLLM (Wu et al., 2025b) employs a logit threshold, where tokens with logits above threshold can be decoded in parallel. By sweeping this threshold, we can adjust the quality–speed trade-off and obtain multiple parallelism–accuracy pairs, which can then be used to plot a curve of accuracy vs. parallelism. We refer to this as *accuracy–parallelism curve* (see Figure 1 for an illustration), which characterizes the trade-off frontier that dLLMs navigate.

A natural first attempt is to summarize the curve by the area under the curve (AUC). However, plain AUC has a serious failure mode: it can reward models that achieve high speed by allowing accuracy to collapse. The right side of the curve can contribute a substantial area even if the model is no longer useful in practice. We therefore require a metric that strongly favors remaining in a high-accuracy regime and only then rewards higher parallelism.

To this end, we propose AUP (*Accuracy Under Parallelism*) as a weighted area under the accuracy–parallelism curve, where the weight penalizes accuracy drops relative to the best achievable accuracy on that task. Formally, let $\mathcal{S} = \{(\rho_i, y_i)\}_{i=1}^m$ be a set of parallelism-accuracy pairs, where $\rho_1 < \rho_2 < \ldots < \rho_m$, $\rho_i \in \mathbb{R}^+$ denotes the parallelism (measured by TPF), and $y_i \in [0, 100]$ represents accuracy in percentage. We define a minimum accuracy threshold $y_{\min} = y_1 - 5$ to avoid measuring in regimes of significant accuracy degradation. Only points satisfying $y_i \geq y_{\min}$ are included. AUP is then defined as:

$$\text{AUP} \triangleq \rho_1 y_1 + \sum_{i=2}^m (\rho_i - \rho_{i-1}) \left( \frac{y_i W(y_i) + y_{i-1} W(y_{i-1})}{2} \right),$$

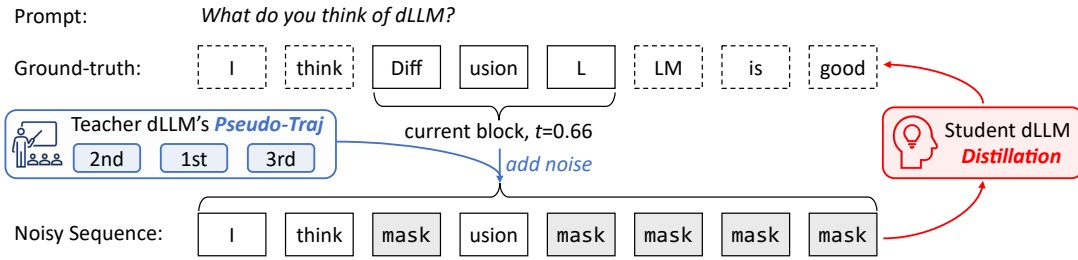

**Figure 2.** Illustration of the trajectory-based distillation recipe in d3LLM, where we combine the pseudo-trajectory of the teacher dLLM model and the ground-truth prompt-response pair to construct a noisy sequence for training the student dLLM.

where the weighting function is defined as $W(y) = \min(e^{-\alpha(1-y/y_{\max})}, 1)$, with a penalty factor $\alpha$ and $y_{\max}$ denotes the highest accuracy achieved on that task. This weight penalizes lower-accuracy regions to emphasize both high parallelism and stable performance, as illustrated in the shaded area in Figure 1.

The intuition behind AUP is simple: (i) If parallelism is increased without losing accuracy, AUP increases a lot. (ii) Instead, if parallelism is increased by sacrificing accuracy, AUP increases only a little, as the penalty suppresses low-accuracy regimes. AUP thus provides a unified measure of quality under increasing parallelism, encouraging to achieve a fair balance between accuracy and parallelism.

**Remark 1** (Choice of $\alpha$). The hyperparameter $\alpha$ controls the penalty for accuracy degradation. A larger $\alpha$ increases sensitivity to performance drops, causing the contribution of parallelism gains to decay exponentially with increasing error rate. In the ideal case where a method improves parallelism without compromising accuracy, AUP reduces to the standard area under the parallelism-accuracy curve (AUC). We set $\alpha = 3$ as the default, as it provides a reasonable balance between parallelism and accuracy. We validate different choices of $\alpha$ in Appendix A.4.

**Remark 2** (Why Not Optimize AUP Directly?). Although AUP effectively captures the accuracy–parallelism trade-off, it cannot serve as a training objective because it is non-differentiable. Specifically, the parallelism $\rho_i$ measured by TPF is a runtime statistic from the decoding process rather than an explicit function of model parameters $\theta$. Moreover, AUP is defined over a set of parallelism-accuracy pairs $\{(\rho_i, y_i)\}_{i=1}^{m}$, each corresponding to a discrete hard confidence threshold selected for decoding, which precludes gradient computation. We therefore treat AUP as an evaluation metric. Instead, inspired by this accuracy–parallelism trade-off, we propose d3LLM as described in Section 3.

## 3. d3LLM: Balance Accuracy and Parallelism

This section introduces our approach, d3LLM framework.

**Motivation.** As demonstrated in the previous section,

dLLMs inherently face an accuracy–parallelism trade-off. While AUP cannot be directly optimized as a training objective (see Appendix A.4), it provides a guiding principle: balancing accuracy and parallelism is essential. Based on this insight, we propose d3LLM (*pseuDo-Distilled-Diffusion Large Language Model*), a framework that improves both training and inference recipes:

(i) *Pseudo-trajectory distillation (training):* We incorporate the teacher dLLM's own decoding trajectory into the distillation process to provide intermediate supervision indicating which tokens can be safely decoded earlier, thereby *improving parallelism*. We further employ a curriculum learning strategy that progressively increases difficulty, enabling stable training and *preserving accuracy*.

(ii) *Multi-block decoding with KV-refresh (inference):* We introduce entropy-based multi-block decoding that simultaneously decodes multiple blocks by prioritizing high-confidence tokens, thereby *improving parallelism*. To mitigate quality degradation, we employ a KV-cache refresh mechanism that periodically recomputes all previously cached states, thereby *preserving accuracy*.

### 3.1. Pseudo-Trajectory-Based Distillation Recipe

We propose a novel *pseudo-trajectory-based distillation* recipe. Specifically, it incorporates following key technique:

**Utilizing the Teacher dLLM's Pseudo-Trajectory.** A key challenge in distillation is that dLLM's intermediate supervision is unavailable: we usually only have prompt–response pairs, without teacher's intermediate states. Ideally, when the teacher's output matches the ground truth, its decoding trajectory provides an ideal *real-trajectory* for teaching the student the correct generation order, but such cases are rare. To overcome this, we instead use the teacher's own decoding trajectory as a *pseudo-trajectory*, even when its final answer differs from the ground truth.

Specifically, given a prompt $\mathbf{x}$ and a predefined maximum output length $n$, we first use the teacher dLLM to generate and record its own decoding trajectory $\{\mathcal{T}_1, \ldots, \mathcal{T}_n\}$, where $\mathcal{T}_i \in \mathbb{R}^n$, $\forall i \in \{1, \ldots, n\}$. Here, we constrain the

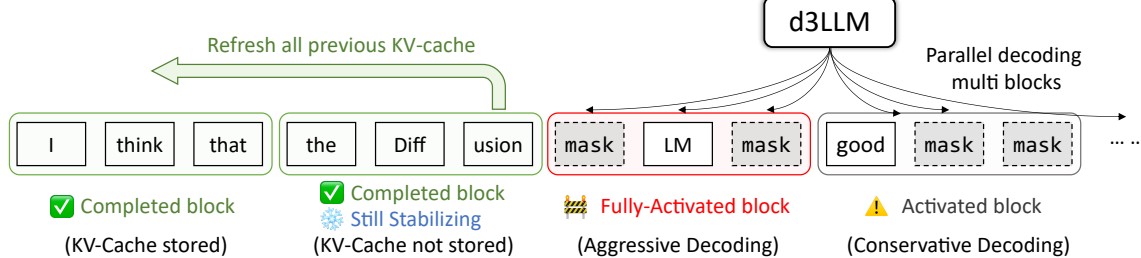

**Figure 3.** Illustration of the multi-block decoding strategy in d3LLM, where we decode multiple blocks in parallel based on token entropy, and we introduce a KV-cache together with a KV-refresh mechanism to mitigate quality degradation.

teacher model to unmask exactly one token at each decoding step, and we continue generation beyond the EOS token so that the output length is exactly $n$. Rather than relying on the content of the teacher's response, we extract only the order in which tokens are decoded, which we refer to as the *pseudo-trajectory* of the teacher. We combine it with the ground-truth prompt–response pair $(\mathbf{x}, \mathbf{y})$ and construct a *noisy sequence* $\widetilde{\mathbf{y}} \in \mathbb{R}^n$ that simulates teacher's intermediate state during the decoding. Formally, let $t \in [0, 1]$ denote the mask ratio, and $w = \{s, \ldots, s + k\}$ be a decoding window of length $k$, the noisy sequence $\widetilde{\mathbf{y}} \in \mathbb{R}^n$ is

$$[\widetilde{\mathbf{y}}]_i = \begin{cases} [\mathbf{y}]_i & \text{if } i \leq s \text{ or } \left[\mathcal{T}_{s+\lceil kt \rceil}\right]_i \neq \texttt{mask}, \\ \texttt{mask} & \text{if } i > s + k \text{ or } \left[\mathcal{T}_{s+\lceil kt \rceil}\right]_i = \texttt{mask}, \end{cases}$$

where $\texttt{mask}$ is the special mask token ID, and $[\cdot]_i$ denotes the $i$-th token in the trajectory sequence. By training the student dLLM on this noisy input by requiring it to predict the ground-truth labels of the masked tokens (using the cross-entropy loss function), the model learns to unmask tokens in an order aligned with the teacher's decoding order. This leads to smoother and more efficient token generation, yielding a *18% improvement in TPF* compared to strategies that use random masking.

**Curriculum Noise Level.** To preserve accuracy during distillation, we introduce a *progressive noise schedule* by gradually increasing the mask ratio $t$ from 0.0 to 0.8 during the training process. This curriculum learning approach encourages the model to learn from easier to harder decoding scenarios, thereby enhancing its robustness and decoding efficiency while maintaining generation quality. Empirically, this strategy further improves the model's tokens-per-forward (TPF) by approximately *12%* compared to using a fixed mask ratio. Without this curriculum strategy, we observe that the distillation process becomes unstable and the model is more likely to suffer accuracy degradation.

**Curriculum Window Size.** Inspired by Hu et al. (2025), we also employ a *progressive window size* during training. Instead of fixing the decoding window length $k$, we gradually increase it from 16 to 32 during the training process. This allows the model to adapt to increasingly larger context

spans, facilitating a smoother distillation process and stable token generation. This approach leads to an additional *8% improvement in TPF* compared to a constant window size.

### 3.2. Multi-Block Decoding Strategy

In addition to the novel distillation recipe, we also introduce a parallel decoding strategy, designed to maximize parallelism across multiple blocks, and preserving the accuracy.

**Entropy-Based Multi-Block Parallel Decoding.** Inspired by D2F (Wang et al., 2025a), we propose an *entropy-based multi-block decoding* method. Unlike conventional block diffusion methods, which operate strictly within a single block, our method enables decoding of both the current and future blocks in parallel. We select tokens to decode based on the entropy threshold, in which lower-entropy (more confident) predictions are first to be unmasked.

Each block can be in one of the five states: `Inactive`, `Activated`, `Fully-Activated`, `Completed but Stabilizing`, and `Completed`. Transition rules are as follows: we create a new `Activated` block when its preceding block reaches $10\%$ completion and employ a conservative decoding strategy for this block, generating tokens only when they meet the entropy threshold. When the preceding block reaches $95\%$ completion, the `Activated` block transitions to a `Fully-Activated` state, where a more aggressive strategy is used by decoding at least one token per forward pass, regardless of the threshold. Once all tokens in a block are unmasked, the block enters the `Completed but Stabilizing` state, during which we perform forward passes without using the KV cache and refresh previous caches. After 1 or 2 rounds, the block becomes `Completed`, and we store its KV cache. In addition, we apply a periodic-refresh strategy that updates the KV cache every few rounds. This multi-block decoding strategy increases TPF by *30%*, and the KV-cache refresh mechanism helps maintain the accuracy.

**KV-Cache and KV-Refresh Mechanism.** To further improve decoding throughput while maintaining generation quality, particularly in long-context settings, we incorporate

a *KV-cache* mechanism alongside a periodic *KV-refresh*. Specifically, after completing each block, we introduce a short delay before caching its key–value states to ensure that the cache remains reliable and does not lead to performance degradation, and we perform full forward passes to refresh KV-caches before the stabilizing block. This hybrid strategy maintains decoding accuracy while significantly improving TPS by approximately *35%* in long-context scenarios.

**Early Stopping on EOS Token.** We implement an *early stopping mechanism* that halts decoding once the end-of-sequence (EOS) token is generated. In standard dLLM decoding, the model continues to perform forward passes until a predetermined number of steps is completed, regardless of whether meaningful content is still being generated. This leads to unnecessary computation, particularly for shorter outputs where the model has already finished generating. Our early stopping mechanism monitors the generated tokens at each decoding step and terminates the process immediately upon detecting the EOS token. This simple yet effective optimization eliminates redundant forward passes and yields a *5% improvement in TPF*.

By combining our distillation recipe with the decoding strategy, our d3LLM framework surpasses previous SOTA dLLM methods in efficiency without sacrificing accuracy, thus striking a balance between accuracy and parallelism.

# 4. Experiments

In this section, we present the empirical evaluations. We aim to answer the following research questions:

**Q1.** Is the metric AUP reasonable?

**Q2.** How effective is d3LLM in terms of AUP score?

**Q3.** Is each module in d3LLM effective?

## 4.1. Experimental Setup

We first introduce the experimental setup as follows, including the contenders, implementation details, and datasets.

**Contenders.** To validate the effectiveness of our approach, we compare d3LLM framework with state-of-the-art dLLM methods, including vanilla LLaDA (Nie et al., 2025) and Dream (Ye et al., 2025), training-free inference acceleration method Fast-dLLM (Wu et al., 2025b), causal block-diffusion methods Fast-dLLM-v2 (Wu et al., 2025a), dParallel (Chen et al., 2025), and D2F (Wang et al., 2025a).

**Implementation Details.** Our experiments are conducted on three foundational diffusion models: LLaDA (Nie et al., 2025), Dream (Ye et al., 2025), and Dream-Coder (Xie et al., 2025). From these, we derive three models, *d3LLM-LLaDA*, *d3LLM-Dream*, and *d3LLM-Coder*, each trained using the same trajectory-based distillation recipe and multi-block

decoding strategy outlined previously. When inference, we use a single GPU and fix the batch size to 1 for all models.

**Benchmark Datasets.** We present benchmark results across five representative tasks: GSM8K-CoT (chain-of-thought reasoning) (Gao et al., 2024), MATH (mathematical problem solving) (Lewkowycz et al., 2022), HumanEval (code generation) (Chen et al., 2021), MBPP (Python programming) (Austin et al., 2021b), and a long-context math reasoning task (5-shot GSM8K (Cobbe et al., 2021), with a prompt length $\approx 1000$). These datasets span diverse domains and problem types and are widely used in the research community. In addition, their relatively long output lengths allow us to effectively evaluate the models' parallel decoding capabilities together with their accuracy. We assess the performance of different dLLM methods using three key metrics: parallelism, measured by the tokens per forward pass (TPF); accuracy (the solve rate / pass@1 accuracy depending on the benchmark); and our proposed AUP (*Accuracy Under Parallelism*) score.

## 4.2. Evaluation of d3LLM Framework

In this part, we present the evaluation results of d3LLM.

**Results on LLaDA-based Models.** As shown in Table 1, our *d3LLM-LLaDA* consistently achieves the highest AUP scores across all five benchmark tasks, demonstrating the effectiveness of our proposed framework. Specifically, *d3LLM-LLaDA* achieves an AUP score of 637.7 on GSM8K-CoT, significantly outperforming the second-best method dParallel (358.1). On MATH dataset, *d3LLM-LLaDA* achieves 107.6 AUP, which is higher than dParallel (64.5). Similar improvements are observed on HumanEval and Long-GSM8K. This superior performance can be attributed to d3LLM's pseudo-trajectory distillation, which enables the model to learn the teacher's token unmasking order, and the multi-block decoding strategy, which maximizes parallelism while maintaining accuracy through the KV-cache refresh mechanism.

These results also validate the reliability of our AUP metric (**Q1**). For example, on the MBPP dataset with the LLaDA-based model, although many methods achieve parallelism (TPF) greater than 1, their accuracy degradation compared with the best-performing model (Qwen-2.5-7B-it) is substantial, leading to low overall utility. This demonstrates that AUP metric faithfully captures the accuracy–parallelism trade-off: methods that sacrifice accuracy for parallelism are penalized, while those that maintain accuracy while improving parallelism are rewarded.

Figure 4(a) visualizes the accuracy–parallelism curve on MATH, where our *d3LLM-LLaDA* (red curve) dominates the upper-right region, achieving both higher parallelism and competitive accuracy. The radar chart in Figure 4(b)

**Table 1.** Comparison of *d3LLM-LLaDA* with other LLaDA-based models. We report the TPF (tokens per forward), Accuracy, and AUP (Accuracy Under Parallelism) score together with standard deviations over three runs. The best results are highlighted in **bold**.

| Benchmark | Method | TPF ↑ | Acc (%) ↑ | AUP Score ↑ |
|---|---|---|---|---|
| **GSM8K-CoT** (0-shot) | LLaDA | $1.00 \pm 0.0$ | $72.6 \pm 0.2$ | $72.6 \pm 0.2$ |
| | Fast-dLLM-LLaDA | $2.77 \pm 0.1$ | $74.7 \pm 0.2$ | $205.8 \pm 6.4$ |
| | D2F-LLaDA | $2.88 \pm 0.1$ | $73.2 \pm 0.3$ | $209.7 \pm 6.1$ |
| | dParallel-LLaDA | $5.14 \pm 0.1$ | $72.6 \pm 0.2$ | $358.1 \pm 6.2$ |
| | **d3LLM-LLaDA** | $\mathbf{9.11} \pm \mathbf{0.1}$ | $73.1 \pm 0.3$ | $\mathbf{637.7} \pm \mathbf{6.8}$ |
| **MATH** (4-shot) | LLaDA | $1.00 \pm 0.0$ | $32.2 \pm 0.4$ | $32.2 \pm 0.4$ |
| | Fast-dLLM-LLaDA | $1.97 \pm 0.1$ | $30.8 \pm 0.3$ | $47.2 \pm 2.9$ |
| | D2F-LLaDA | $2.38 \pm 0.1$ | $28.7 \pm 0.2$ | $45.5 \pm 2.8$ |
| | dParallel-LLaDA | $3.17 \pm 0.1$ | $30.2 \pm 0.2$ | $64.5 \pm 3.1$ |
| | **d3LLM-LLaDA** | $\mathbf{5.74} \pm \mathbf{0.1}$ | $30.4 \pm 0.3$ | $\mathbf{107.6} \pm \mathbf{3.2}$ |
| **MBPP** (3-shot) | LLaDA | $1.00 \pm 0.0$ | $41.7 \pm 0.3$ | $41.7 \pm 0.3$ |
| | Fast-dLLM-LLaDA | $2.13 \pm 0.1$ | $38.6 \pm 0.3$ | $56.6 \pm 3.7$ |
| | D2F-LLaDA | $1.94 \pm 0.1$ | $38.0 \pm 0.2$ | $50.0 \pm 3.6$ |
| | dParallel-LLaDA | $2.35 \pm 0.1$ | $40.0 \pm 0.3$ | $60.5 \pm 3.9$ |
| | **d3LLM-LLaDA** | $\mathbf{4.21} \pm \mathbf{0.1}$ | $40.6 \pm 0.2$ | $\mathbf{88.4} \pm \mathbf{4.0}$ |
| **HumanEval** (0-shot) | LLaDA | $1.00 \pm 0.0$ | $38.3 \pm 0.5$ | $38.3 \pm 0.5$ |
| | Fast-dLLM-LLaDA | $2.56 \pm 0.1$ | $37.8 \pm 0.4$ | $54.0 \pm 2.9$ |
| | D2F-LLaDA | $2.69 \pm 0.1$ | $36.6 \pm 0.5$ | $62.0 \pm 2.7$ |
| | dParallel-LLaDA | $4.93 \pm 0.2$ | $39.0 \pm 0.4$ | $83.7 \pm 4.8$ |
| | **d3LLM-LLaDA** | $\mathbf{5.95} \pm \mathbf{0.1}$ | $39.6 \pm 0.6$ | $\mathbf{96.6} \pm \mathbf{3.2}$ |
| **Long-GSM8K** (5-shot) | LLaDA | $1.00 \pm 0.0$ | $78.6 \pm 0.2$ | $78.6 \pm 0.2$ |
| | Fast-dLLM-LLaDA | $2.45 \pm 0.1$ | $78.0 \pm 0.3$ | $175.4 \pm 6.4$ |
| | D2F-LLaDA | $2.70 \pm 0.1$ | $73.7 \pm 0.2$ | $168.5 \pm 6.0$ |
| | dParallel-LLaDA | $4.49 \pm 0.1$ | $76.7 \pm 0.3$ | $309.1 \pm 6.2$ |
| | **d3LLM-LLaDA** | $\mathbf{6.95} \pm \mathbf{0.1}$ | $74.2 \pm 0.3$ | $\mathbf{441.1} \pm \mathbf{6.5}$ |

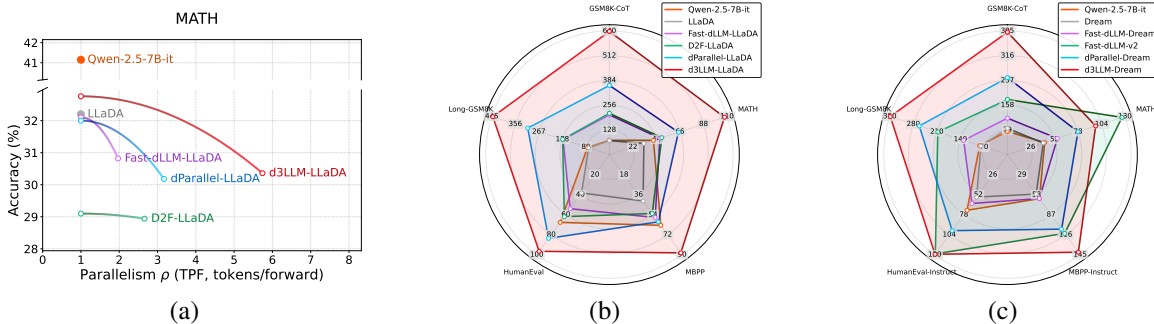

**Figure 4.** (a): Accuracy–parallelism curves of LLaDA-based models on MATH dataset. (b) & (c): Radar charts of AUP score.

further illustrates that *d3LLM-LLaDA* achieves the largest coverage area across all five tasks, indicating its consistent superiority in balancing accuracy and parallelism.

**Results on Dream-based Models.** As shown in Table 2, *d3LLM-Dream* achieves the highest AUP scores on 4 out of 5 tasks, further validating the generalizability of our framework. For example, on GSM8K-CoT, *d3LLM-Dream* achieves an AUP of 391.3, outperforming *dParallel* (245.7) and Fast-dLLM-v2 (156.0). On MBPP-Instruct, *d3LLM-Dream* achieves an AUP score of 141.4, which is higher than *dParallel* (108.0). Similar improvements are observed on other datasets, and the radar chart in Figure 4(c) further demonstrates that *d3LLM-Dream* achieves the largest overall coverage area among Dream-based methods, indicating its balanced superiority across diverse benchmarks. These

results affirmatively answer **Q2**: our d3LLM framework is effective across different base models and tasks.

Notably, on the MATH dataset, *Fast-dLLM-v2* achieves the highest AUP score (126.7), which attributes to its notably higher accuracy (48.7%) compared to other Dream-based methods. We suspect that this stems from the fact that *Fast-dLLM-v2* is finetuned directly from Qwen-2.5-7B with an additional 1B tokens (i.e., the LLaMA–Nemotron post-training dataset (Bercovich et al., 2025)). In contrast, our *d3LLM-Dream* is distilled based on the vanilla Dream and uses only 60M additional tokens. Despite this data disadvantage, *d3LLM-Dream* still achieves competitive performance and the best results on the majority of tasks.

**Wall-Clock Speed Comparison.** In addition to the AUP

**Table 2.** Comparison of *d3LLM-Dream* with other Dream-based models. We report the TPF (tokens per forward), Accuracy, and AUP (Accuracy Under Parallelism) score together with standard deviations over three runs. The best results are highlighted in **bold**.

| Benchmark | Method | TPF ↑ | Acc (%) ↑ | AUP Score ↑ |
|---|---|---|---|---|
| **GSM8K-CoT** (0-shot) | Dream | 1.00 ± 0.0 | 83.9 ± 0.2 | 83.9 ± 0.2 |
| | Fast-dLLM-Dream | 1.44 ± 0.1 | 79.0 ± 0.3 | 116.5 ± 6.3 |
| | Fast-dLLM-v2 | 2.21 ± 0.2 | 77.5 ± 0.2 | 156.0 ± 9.2 |
| | dParallel-Dream | 3.02 ± 0.1 | 82.1 ± 0.3 | 245.7 ± 8.3 |
| | **d3LLM-Dream** | **4.94 ± 0.1** | 81.4 ± 0.3 | **391.3 ± 7.9** |
| **MATH** (4-shot) | Dream | 1.00 ± 0.0 | 39.6 ± 0.2 | 39.6 ± 0.2 |
| | Fast-dLLM-Dream | 1.78 ± 0.1 | 38.3 ± 0.2 | 55.2 ± 3.4 |
| | Fast-dLLM-v2 | 2.61 ± 0.1 | 48.7 ± 0.2 | **126.7 ± 4.2** |
| | dParallel-Dream | 2.94 ± 0.1 | 38.7 ± 0.3 | 77.9 ± 3.5 |
| | **d3LLM-Dream** | **3.92 ± 0.1** | 38.2 ± 0.3 | 97.5 ± 3.8 |
| **MBPP-Instruct** (0-shot) | Dream | 1.00 ± 0.0 | 57.2 ± 0.2 | 57.2 ± 0.2 |
| | Fast-dLLM-Dream | 1.20 ± 0.1 | 53.2 ± 0.3 | 63.6 ± 5.8 |
| | Fast-dLLM-v2 | 2.04 ± 0.1 | 50.1 ± 0.3 | 81.9 ± 5.3 |
| | dParallel-Dream | 2.24 ± 0.1 | 55.4 ± 0.4 | 108.0 ± 5.9 |
| | **d3LLM-Dream** | **2.96 ± 0.2** | 55.6 ± 0.2 | **141.4 ± 8.7** |
| **HumanEval-Instruct** (0-shot) | Dream | 1.00 ± 0.0 | 55.2 ± 0.1 | 55.2 ± 0.1 |
| | Fast-dLLM-Dream | 1.33 ± 0.0 | 54.3 ± 0.3 | 63.5 ± 0.4 |
| | Fast-dLLM-v2 | 2.58 ± 0.1 | 61.7 ± 0.2 | 128.9 ± 5.9 |
| | dParallel-Dream | 2.57 ± 0.2 | 54.3 ± 0.3 | 98.8 ± 9.2 |
| | **d3LLM-Dream** | **3.20 ± 0.1** | 57.1 ± 0.4 | **129.5 ± 6.3** |
| **Long-GSM8K** (5-shot) | Dream | 1.00 ± 0.0 | 79.0 ± 0.3 | 79.0 ± 0.3 |
| | Fast-dLLM-Dream | 1.79 ± 0.1 | 76.6 ± 0.2 | 130.4 ± 6.1 |
| | Fast-dLLM-v2 | 2.58 ± 0.1 | 81.0 ± 0.2 | 207.2 ± 8.5 |
| | dParallel-Dream | 3.49 ± 0.1 | 78.6 ± 0.3 | 262.4 ± 7.3 |
| | **d3LLM-Dream** | **4.80 ± 0.1** | 77.2 ± 0.3 | **348.6 ± 7.9** |

**Table 3.** Throughput comparison of *d3LLM-LLaDA* with contenders on GSM8K-CoT. We report tokens per second (TPS) on H100 and A100 GPUs, and accuracy. Speedup ratios relative to AR model (Qwen-2.5-7B-it) are shown in parentheses.

| Method | H100 TPS ↑ | A100 TPS ↑ | Acc (%) ↑ |
|---|---|---|---|
| Qwen-2.5-7B-it | 57.3 (1.0×) | 50.4 (1.0×) | 74.1 |
| LLaDA | 27.9 (0.5×) | 19.2 (0.4×) | 72.6 |
| Fast-dLLM-LLaDA | 114.3 (2.0×) | 79.1 (1.6×) | 74.7 |
| D2F-LLaDA | 102.1 (1.8×) | 76.2 (1.5×) | 73.2 |
| dParallel-LLaDA | 172.2 (3.0×) | 105.9 (2.1×) | 72.6 |
| **d3LLM-LLaDA** | **288.9 (5.0×)** | **183.3 (3.6×)** | 73.1 |

**Table 4.** Throughput comparison of *d3LLM-Dream* with contenders on GSM8K-CoT. We report tokens per second (TPS) on H100 and A100 GPUs, and accuracy. Speedup ratios relative to AR model (Qwen-2.5-7B-it) are shown in parentheses.

| Method | H100 TPS ↑ | A100 TPS ↑ | Acc (%) ↑ |
|---|---|---|---|
| Qwen-2.5-7B-it | 57.3 (1.0×) | 50.4 (1.0×) | 74.1 |
| Dream | 27.6 (0.5×) | 8.3 (0.2×) | 83.9 |
| Fast-dLLM-Dream | 77.3 (1.3×) | 51.6 (1.0×) | 79.0 |
| Fast-dLLM-v2 | 150.0 (2.6×) | 109.7 (2.2×) | 77.5 |
| dParallel-Dream | 168.4 (2.9×) | 80.2 (1.6×) | 82.1 |
| **d3LLM-Dream** | **235.3 (4.1×)** | **128.2 (2.5×)** | 81.4 |

scores, we further evaluate different methods on multiple hardware platforms, including H100 and A100 GPUs, to measure their wall-clock throughput (measured by tokens per second, TPS). As shown in Table 3, for LLaDA-based models on GSM8K-CoT, our *d3LLM-LLaDA* achieves 288.9 TPS on H100 (5.0× speedup over Qwen-2.5-7B-it) and 183.3 TPS on A100 (3.6× speedup), while maintaining competitive accuracy. Compared to vanilla LLaDA (27.9 TPS on H100), *d3LLM-LLaDA* achieves a remarkable 10.3× speedup. Similarly, as shown in Table 4, our *d3LLM-Dream* achieves 235.3 TPS on H100 GPU (4.1× speedup) and 128.2 TPS on A100 (2.5× speedup), representing an 8.5× improvement over the vanilla Dream (27.6 TPS on H100) while still preserving high accuracy.

To summarize, d3LLM achieves the highest AUP scores

with negligible performance degradation, successfully striking a balance between accuracy and parallelism. It delivers up to 5× speedup over autoregressive decoding (Qwen-2.5-7B-it) on H100 GPUs and approximately 3.6× speedup on A100 GPUs. Due to page limit, we defer more experimental results of about *d3LLM-Coder* to Appendix A.

### 4.3. Ablation Study

In this part, we present the ablation study of our d3LLM.

**Ablation Study on Distillation Recipe.** To validate each component's contribution in the distillation recipe, we conduct ablation studies on *d3LLM-LLaDA* and evaluate it on the GSM8K-CoT dataset. As shown in Table 5 (upper part), we compare our full method with three variants:

**Table 5.** Ablation study on different distillation and decoding strategies of our method. We report the TPF, Accuracy, and AUP score of our *d3LLM-LLaDA* on GSM8K-CoT dataset (0-shot).

| Distillation Recipe | | | Decoding Method | | GSM8K-CoT (0-shot) | | |
|---|---|---|---|---|---|---|---|
| Pseudo-trajectory | Curriculum Noise | Curriculum Window | Multi-block Decoding | Early Stop | TPF ↑ | Acc (%) ↑ | AUP Score ↑ |
| | | | ✓ | ✓ | $6.41 \pm 0.1$ | $72.2 \pm 0.3$ | $441.4 \pm 3.2$ |
| ✓ | | | ✓ | ✓ | $7.55 \pm 0.1$ | $72.1 \pm 0.2$ | $517.7 \pm 3.9$ |
| ✓ | ✓ | | ✓ | ✓ | $8.46 \pm 0.2$ | $69.8 \pm 0.4$ | $551.3 \pm 7.8$ |
| ✓ | ✓ | ✓ | ✓ | ✓ | $\mathbf{9.11} \pm \mathbf{0.1}$ | $\mathbf{73.1} \pm \mathbf{0.3}$ | $\mathbf{637.7} \pm \mathbf{6.8}$ |
| ✓ | ✓ | ✓ | | | $7.01 \pm 0.1$ | $73.2 \pm 0.1$ | $492.9 \pm 4.3$ |
| ✓ | ✓ | ✓ | ✓ | | $9.07 \pm 0.1$ | $73.1 \pm 0.3$ | $635.0 \pm 6.7$ |
| ✓ | ✓ | ✓ | ✓ | ✓ | $\mathbf{9.11} \pm \mathbf{0.1}$ | $\mathbf{73.1} \pm \mathbf{0.3}$ | $\mathbf{637.7} \pm \mathbf{6.8}$ |

**Table 6.** Hyperparameter analysis of *Curriculum Noise Level*. We report the TPF, Accuracy, and AUP score on GSM8K-CoT dataset of *d3LLM-LLaDA* model.

| Noise | TPF ↑ | Acc (%) ↑ | AUP Score ↑ |
|---|---|---|---|
| Fixed ($t$=0.5) | $7.49 \pm 0.1$ | $72.8 \pm 0.5$ | $521.7 \pm 5.4$ |
| Curriculum $0.2 \rightarrow 0.5$ | $8.03 \pm 0.2$ | $72.7 \pm 0.5$ | $557.7 \pm 5.8$ |
| Curriculum $0.0 \rightarrow 0.5$ | $8.85 \pm 0.1$ | $72.9 \pm 0.5$ | $616.9 \pm 6.5$ |
| Curriculum $0.0 \rightarrow 0.8$ | $\mathbf{9.11} \pm \mathbf{0.1}$ | $\mathbf{73.1} \pm \mathbf{0.3}$ | $\mathbf{637.7} \pm \mathbf{6.8}$ |

**Table 7.** Hyperparameter analysis of *Curriculum Window Size*. We report the TPF, Accuracy, and AUP score on GSM8K-CoT dataset of *d3LLM-LLaDA* model.

| Size | TPF ↑ | Acc (%) ↑ | AUP Score ↑ |
|---|---|---|---|
| Fixed ($k$=32) | $8.22 \pm 0.2$ | $69.8 \pm 0.5$ | $536.0 \pm 7.8$ |
| Curriculum $0 \rightarrow 32$ | $8.67 \pm 0.2$ | $72.8 \pm 0.3$ | $603.1 \pm 9.1$ |
| Curriculum $16 \rightarrow 32$ | $\mathbf{9.11} \pm \mathbf{0.1}$ | $\mathbf{73.1} \pm \mathbf{0.3}$ | $\mathbf{637.7} \pm \mathbf{6.8}$ |
| Curriculum $24 \rightarrow 32$ | $8.58 \pm 0.2$ | $71.9 \pm 0.4$ | $584.9 \pm 8.9$ |

(i) a baseline model with only multi-block decoding and early stopping but without any distillation components, (ii) a model with pseudo-trajectory distillation only, and (iii) only without curriculum windows. The baseline without distillation achieves a TPF of 6.41 with 72.2% accuracy. Adding pseudo-trajectory distillation improves TPF by 17.8% (from 6.41 to 7.55) while maintaining similar accuracy, demonstrating that learning the teacher's token unmasking order effectively improves parallelism. Incorporating curriculum noise further increases TPF to 8.46, though with an accuracy drop to 69.8%, indicating that the curriculum learning strategy enables more aggressive parallel decoding. Finally, adding curriculum window size yields our full model with TPF of 9.11 and accuracy of 73.1%, achieving a TPF improvement while recovering accuracy. This demonstrates that the curriculum window strategy not only improves parallelism but also stabilizes the distillation process.

**Ablation Study on Decoding Strategy.** As shown in Table 5 (lower part), we further ablate the decoding strategy components on *d3LLM-LLaDA*. Starting from our full distillation recipe, we compare: (i) vanilla block diffusion decoding without multi-block or early stopping, (ii) multi-block decoding without early stopping, and (iii) our complete decoding strategy with both multi-block decoding and early stopping. The multi-block decoding strategy enables parallel decoding across multiple blocks based on token entropy, which significantly improves TPF by allowing confident tokens in future blocks to be decoded simultaneously with the current block. The early stopping mechanism further optimizes efficiency by terminating decoding upon generating the EOS token, eliminating redundant forward passes. To-

gether, these components achieve a TPF of 9.11 with 73.1% accuracy, yielding the highest AUP score.

These ablation results affirmatively answer **Q3**: each component in d3LLM contributes meaningfully to the overall performance. The pseudo-trajectory distillation provides intermediate supervision for learning efficient token unmasking orders, the curriculum noise and window strategies enable stable curriculum learning, and the multi-block decoding with early stopping maximizes inference-time parallelism. The synergy of these components enables d3LLM to achieve the best balance between accuracy and parallelism.

**Hyperparameter Analysis.** We further investigate the impact of key hyperparameters in our curriculum learning strategy: the *curriculum noise level* and the *curriculum window size* strategy. As shown in Table 6, using a fixed noise level ($t = 0.5$) achieves a TPF of 7.49 and accuracy of 72.8%, while our curriculum noise strategy ($0.0 \rightarrow 0.8$) improves TPF to 9.11 with accuracy of 73.1%, yielding a 21.6% improvement in TPF and 22.2% improvement in AUP score. This validates that gradually increasing the noise level during training enables the model to first learn basic token dependencies before handling more challenging masking patterns. Similarly, Table 7 shows that a fixed window size ($k$=32) achieves a TPF of 8.22 with accuracy of 69.8%, while our curriculum window strategy ($16 \rightarrow 32$) improves TPF to 9.11 with accuracy of 73.1%, yielding a 19.0% improvement in AUP score. Interestingly, starting from too small a window ($0 \rightarrow 32$) leads to lower accuracy and AUP score, suggesting that an appropriate window size is crucial for stable distillation. These results demonstrate that our curriculum learning strategy with properly tuned

**Table 8.** Analysis of weighting functions in AUP on GSM8K-CoT using LLaDA-based models. We evaluate exponential (Exp), power (Pow), and linear (Lin) penalty functions with varying $\alpha$. Rankings are shown in parentheses.

| Method | Exponential | | | Power | | | Linear | | |
|---|---|---|---|---|---|---|---|---|---|
| | $\alpha=1$ | $\alpha=3$ | $\alpha=5$ | $\alpha=1$ | $\alpha=3$ | $\alpha=5$ | $\alpha=1$ | $\alpha=5$ | $\alpha=10$ |
| LLaDA | 72.6 (#5) | 72.6 (#5) | 72.6 (#5) | 72.6 (#5) | 72.6 (#5) | 72.6 (#5) | 72.6 (#5) | 72.6 (#5) | 72.6 (#5) |
| Fast-dLLM-LLaDA | 206.6 (#4) | 205.8 (#4) | 204.9 (#4) | 206.6 (#4) | 205.8 (#4) | 204.9 (#4) | 206.6 (#4) | 204.9 (#3) | 202.7 (#3) |
| D2F-LLaDA | 210.8 (#3) | 209.7 (#3) | 208.7 (#3) | 210.8 (#3) | 209.8 (#3) | 208.7 (#3) | 210.8 (#3) | 208.7 (#4) | 205.4 (#4) |
| dParallel-LLaDA | 370.9 (#2) | 358.1 (#2) | 346.0 (#2) | 370.8 (#2) | 357.9 (#2) | 345.6 (#2) | 370.8 (#2) | 344.0 (#2) | 310.4 (#2) |
| **d3LLM-LLaDA** | **659.4** (#1) | **637.7** (#1) | **616.8** (#1) | **659.2** (#1) | **637.3** (#1) | **616.3** (#1) | **659.2** (#1) | **614.0** (#1) | **557.4** (#1) |

schedules effectively balances parallelism and accuracy.

**Analysis of Weighting Function in AUP.** In previous experiments, we adopt an exponential weighting function $W(y) = \min\big(\exp\big(-\alpha \cdot (1 - y/y_{\max})\big), 1\big)$ in the AUP metric, as the exponential form naturally amplifies the penalty for accuracy degradation. To verify that the comparative conclusions are not sensitive to this choice, we evaluate two additional penalty functions: a power form $W(y) = (y/y_{\max})^\alpha$ and a linear form $W(y) = \max\big(1 - \alpha \cdot (1 - y/y_{\max}), 0\big)$, each with multiple $\alpha$ values. The results for LLaDA-based methods on GSM8K-CoT are reported in Table 8. Across all nine configurations spanning three functional families, the ranking of methods remains consistent, with d3LLM achieving the highest AUP in every case. This confirms that the AUP metric is robust to the choice of weighting function, and that our conclusions are not artifacts of a particular penalty design.

## 5. Conclusion

In this paper, we observe a fundamental accuracy–parallelism trade-off in dLLMs. Existing methods typically focus on only one side of the coin, targeting either efficiency or performance. Moreover, previous training approaches with random masking provide no guidance on which tokens can be safely decoded early, while previous multi-block decoding methods degrade quality due to incomplete or erroneous context. To this end, we propose d3LLM (*pseuDo-Distilled-Diffusion LLM*) framework, striking a balance between the accuracy and the parallelism: at training time, *pseudo-trajectory distillation* teaches the model which tokens can be confidently decoded early by leveraging the teacher's unmasking order; at inference time, *entropy-based multi-block decoding* with a *KV-cache refresh mechanism* enables high parallelism while maintaining accuracy. To better evaluate dLLMs, we also introduce AUP (*Accuracy Under Parallelism*), a new metric that jointly measures accuracy and parallelism. Experiments on LLaDA, Dream, and Dream-Coder, demonstrate that d3LLM achieves the highest AUP score on 9 out of 10 benchmark tasks, delivering up to $3.6\times$–$5\times$ speedup over AR models depending on the GPU platform, and $10\times$ speedup over vanilla LLaDA/Dream, with negligible accuracy degradation.

## Acknowledgements

Yu-Yang Qian and Peng Zhao were supported by NSFC (62576164) and the "111 Center" (No. B26023), and the Fundamental Research Funds for the Central Universities (2026300331). Junda Su, Lanxiang Hu, Peiyuan Zhang, and Hao Zhang were supported by UCSD HDSI. The computing resources were provided by MBZUAI IFM and Nvidia's donation. Part of this work was conducted during Yu-Yang Qian's visit to UCSD. The authors would like to thank Chongxuan Li for the insightful and helpful discussions, and thank Hao-Cong Wu for the assistance in experiments.

## Impact Statement

This paper presents work aimed at advancing the field of diffusion LLMs by tackling the accuracy-parallelism trade-off. Our proposed method enables more efficient inference without much accuracy degradation, which may contribute to reduced computational costs. There are many potential societal consequences of our work, none of which we feel must be specifically highlighted here.

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

# A. Additional Experiments

This section provides additional experimental details for the main paper.

## A.1. Detailed Results of LLaDA-based Models

For the **LLaDA-based models**, we compare our *d3LLM-LLaDA* with *vanilla LLaDA*, *Fast-dLLM-LLaDA*, *D2F*, and *dParallel-LLaDA*. The detailed experimental results of accuracy–parallelism curve and AUP scores are shown below.

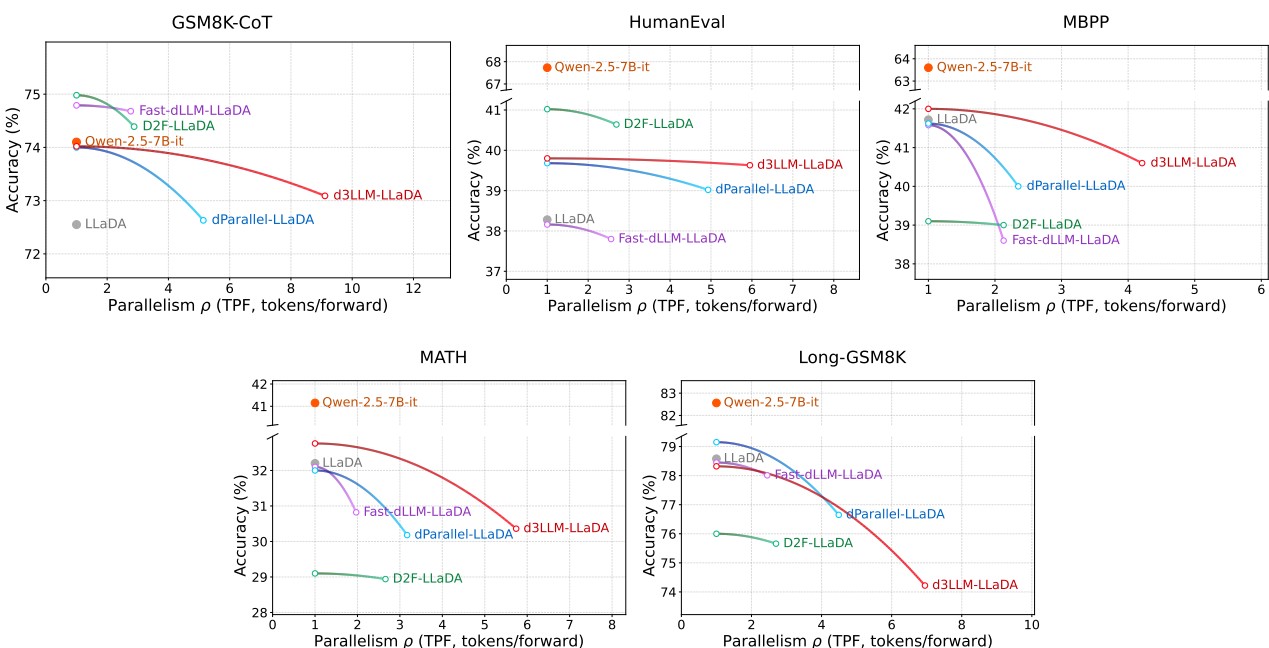

**Figure 5.** Accuracy–parallelism curves for LLaDA-based models across five benchmark tasks (i.e., GSM8K-CoT, HumanEval, MBPP, MATH, and Long-GSM8K).

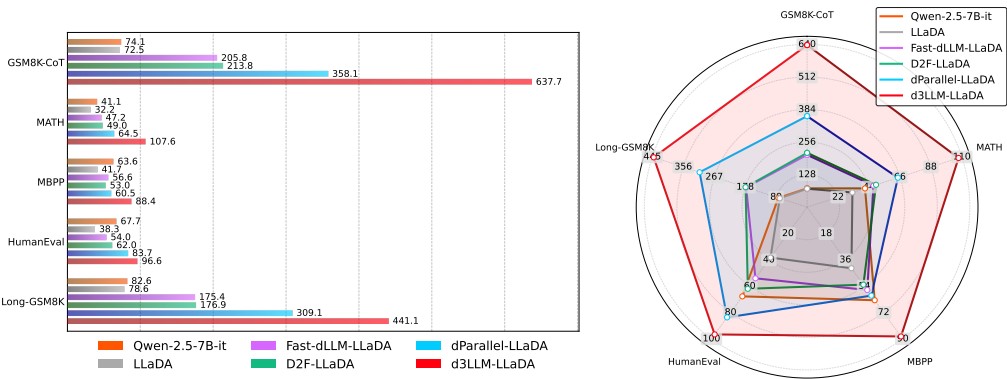

**Figure 6.** AUP scores histogram and radar chart comparing different LLaDA-based methods.

## A.2. Detailed Results of Dream-based Models

For the **Dream-based models**, we compare our *d3LLM-Dream* with *vanilla Dream*, *Fast-dLLM-Dream*, *Fast-dLLM-v2-7B*, and *dParallel-Dream*. The detailed experimental results of accuracy–parallelism curve and AUP scores are shown below.

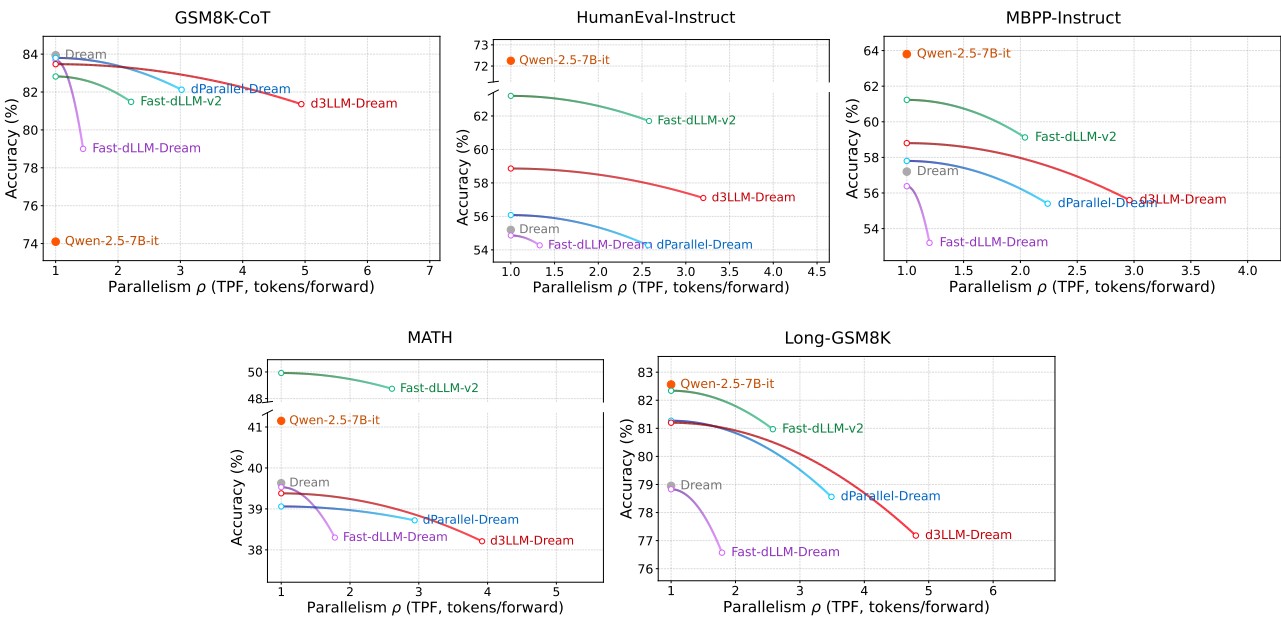

**Figure 7.** Accuracy–parallelism curves for Dream-based models across five benchmark tasks (i.e., GSM8K-CoT, HumanEval-Instruct, MBPP-Instruct, MATH, and Long-GSM8K).

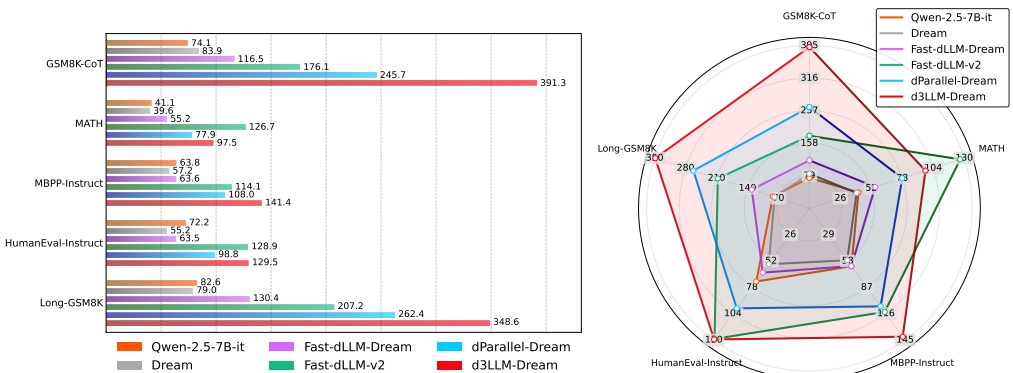

**Figure 8.** AUP scores histogram and radar chart comparing different Dream-based methods.

## A.3. Coder Models

We present the results of our *d3LLM-Coder* on the Coder benchmark tasks.

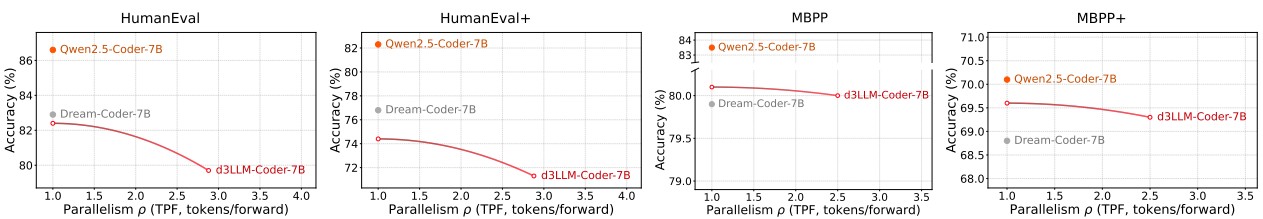

**Figure 9.** Accuracy–parallelism curves for Coder-based models across five benchmark tasks (i.e, GSM8K-CoT, HumanEval, MBPP, MATH, and Long-GSM8K).

**Table 9.** Comparison of *d3LLM-Coder-7B* with contenders. We report the Tokens Per Forward (TPF), Accuracy, and Accuracy Under Parallelism (AUP). The best results are highlighted in **bold**.

| Benchmark | Method | TPF ↑ | Accuracy ↑ | AUP Score ↑ |
|---|---|---|---|---|
| **HumanEval** (0-shot) | Qwen2.5-Coder-7B | 1.00 | 86.6 | 86.6 |
| | Dream-Coder-7B | 1.00 | 82.9 | 82.9 |
| | **d3LLM-Coder-7B** | **2.88** | 79.7 | **208.4** |
| **HumanEval+** (0-shot) | Qwen2.5-Coder-7B | 1.00 | 82.3 | 82.3 |
| | Dream-Coder-7B | 1.00 | 76.8 | 76.8 |
| | **d3LLM-Coder-7B** | **2.88** | 71.3 | **171.7** |
| **MBPP** (0-shot) | Qwen2.5-Coder-7B | 1.00 | 83.5 | 83.5 |
| | Dream-Coder-7B | 1.00 | 79.9 | 79.9 |
| | **d3LLM-Coder-7B** | **2.50** | 80.1 | **186.4** |
| **MBPP+** (0-shot) | Qwen2.5-Coder-7B | 1.00 | 70.1 | 70.1 |
| | Dream-Coder-7B | 1.00 | 68.8 | 68.8 |
| | **d3LLM-Coder-7B** | **2.50** | 69.3 | **170.9** |

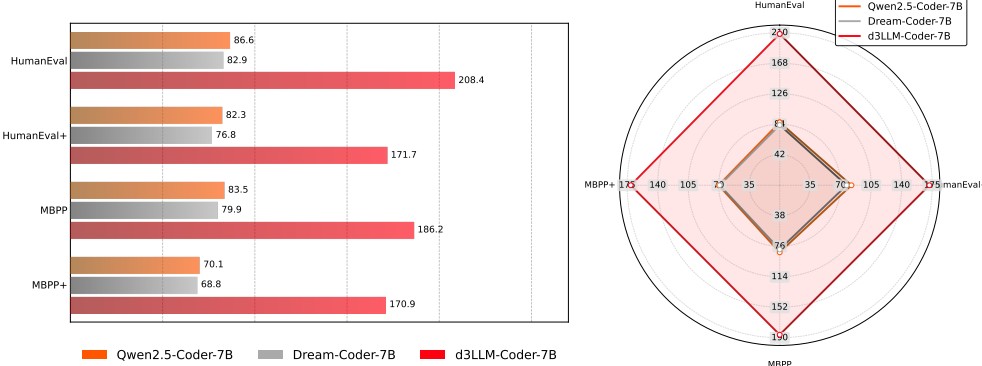

**Figure 10.** AUP scores histogram and radar chart comparing different Coder-based methods.

## A.4. Validation of the Hyperparameter in AUP Metric

In this section, we validate different choices of hyperparameter $\alpha$ as defined in the AUP (*Accuracy Under Parallelism*) metric. Recall that $\alpha$ controls the penalty for accuracy degradation: a larger $\alpha$ increases sensitivity to performance drops, causing the contribution of throughput to decay exponentially with the error rate. We evaluate all methods on GSM8K-CoT with $\alpha \in \{1, 2, 3, 5, 10\}$ and report the AUP scores in Tables 10 and 11.

As shown in the tables, *d3LLM* consistently achieves the highest AUP scores across different $\alpha$ values for both LLaDA-based and Dream-based methods. When $\alpha$ is small (e.g., $\alpha = 1$), the metric is more tolerant to accuracy drops and primarily rewards parallelism gains. As $\alpha$ increases, the penalty for accuracy degradation becomes more severe, and methods with larger accuracy drops see their AUP scores decrease more sharply. Notably, our d3LLM maintains competitive AUP scores under all $\alpha$ values, demonstrating that our approach achieves a favorable balance between parallelism improvement and accuracy preservation. We set $\alpha = 3$ as the default in our main experiments, as it provides a reasonable trade-off between accuracy and parallelism.

**Remark 3** (Hardware-Independence). AUP is *hardware-independent* because it is built on TPF rather than TPS. TPS is heavily affected by hardware generation, kernel fusion, and the inference framework. The same algorithm can appear dramatically different depending on system details. For instance, in our experiments, our *d3LLM-LLaDA* model (introduced in Section 3) achieves approximately $5\times$ higher TPS than an AR baseline (Qwen-2.5-7B-it) on an NVIDIA H100 GPU. However, this advantage shrinks significantly on an NVIDIA A100 GPU. In contrast, TPF captures algorithmic parallelism: how many tokens are generated per forward pass, which is much more stable across hardware. Therefore, AUP gives a fairer view of algorithmic progress, without requiring everyone to run on the exact same GPU or inference engine, helping the community focus on algorithmic design without requiring access to particular GPUs.

**Table 10.** Sensitivity analysis of $\alpha$ on LLaDA-based methods. We report AUP scores with different $\alpha$ values on GSM8K-CoT.

| Method | AUP Score ↑ | | | | |
|---|---|---|---|---|---|
| | $\alpha=1$ | $\alpha=2$ | $\alpha=3$ | $\alpha=5$ | $\alpha=10$ |
| Qwen-2.5-7B-it | 74.1 | 74.1 | 74.1 | 74.1 | 74.1 |
| LLaDA | 72.6 | 72.6 | 72.6 | 72.6 | 72.6 |
| Fast-dLLM-LLaDA | 206.6 | 206.2 | 205.8 | 204.9 | 202.8 |
| D2F-LLaDA | 214.8 | 214.3 | 213.8 | 212.7 | 210.1 |
| dParallel-LLaDA | 370.9 | 364.4 | 358.1 | 346.0 | 318.3 |
| **d3LLM-LLaDA** | **659.4** | **648.4** | **637.7** | **616.8** | **568.4** |

**Table 11.** Sensitivity analysis of $\alpha$ on Dream-based methods. We report AUP scores with different $\alpha$ values on GSM8K-CoT.

| Method | AUP Score ↑ | | | | |
|---|---|---|---|---|---|
| | $\alpha=1$ | $\alpha=2$ | $\alpha=3$ | $\alpha=5$ | $\alpha=10$ |
| Qwen-2.5-7B-it | 74.1 | 74.1 | 74.1 | 74.1 | 74.1 |
| Dream | 83.9 | 83.9 | 83.9 | 83.9 | 83.9 |
| Fast-dLLM-Dream | 118.4 | 117.4 | 116.5 | 114.8 | 111.2 |
| Fast-dLLM-v2 | 180.1 | 178.1 | 176.1 | 172.3 | 163.4 |
| dParallel-Dream | 249.5 | 247.6 | 245.7 | 242.2 | 233.8 |
| **d3LLM-Dream** | **402.4** | **396.8** | **391.3** | **380.8** | **356.8** |

## A.5. Compare with Other Contenders

In this section, we compare our *d3LLM framework* with SDTT (Self-Distillation Through Time) (Deschenaux & Gulcehre, 2025), a distillation method for discrete diffusion models that reduces the number of inference steps by training the student to match the KL divergence of the teacher's multi-step predictions. To ensure a fair comparison, we train *SDTT-LLaDA* under the same configuration as *d3LLM-LLaDA*: both methods use the same distillation dataset, training epochs, effective batch size, optimizer, and learning rate ($2 \times 10^{-5}$ for SDTT, selected as the

**Table 12.** Comparison with SDTT on GSM8K-CoT using LLaDA-based models. We report Acc, TPF, and AUP scores.

| Method | Acc (↑) | TPF (↑) | AUP (↑) |
|---|---|---|---|
| LLaDA | 72.55 | 1.00 | 72.55 |
| SDTT-LLaDA (Bwd KL) | 71.62 | 4.53 | 309.71 |
| SDTT-LLaDA (Fwd KL) | 72.17 | 4.29 | 310.33 |
| **d3LLM-LLaDA** | **73.09** | **9.11** | **637.65** |

best among $\{5e\text{-}6, 2e\text{-}5, 4e\text{-}5, 6e\text{-}5\}$ via grid search). We evaluate SDTT with both the reverse KL (as used in the original paper of Deschenaux & Gulcehre (2025)) and forward KL as the loss function.

The results on GSM8K-CoT are summarized in Table 12. SDTT achieves remarkable speedups over vanilla LLaDA with only marginal accuracy degradation, confirming the general effectiveness of distillation for accelerating dLLMs. However, d3LLM substantially outperforms both SDTT variants in terms of AUP. This advantage stems from two key design choices: (i) *pseudo-trajectory distillation*, which leverages the teacher's own decoding order to provide intermediate supervision on which tokens can be confidently decoded at early steps; and (ii) *curriculum learning strategy*, which progressively increases training difficulty. Together, these strategies yield more refined distillation guidance, enabling d3LLM to achieve a significantly better accuracy–parallelism trade-off.

**Table 13.** Sensitivity analysis of $y_{\min}$ on GSM8K-CoT (LLaDA-based). We report AUP scores under varying $\Delta y$ (where $y_{\min} = y_1 - \Delta y$). Rankings are shown in parentheses.

| Method | $\Delta y=1$ | $\Delta y=2$ | $\Delta y=3$ | $\Delta y=5$ | $\Delta y=8$ | $\Delta y=10$ |
|---|---|---|---|---|---|---|
| Qwen-2.5-7B-it | 74.1 (#6) | 74.1 (#6) | 74.1 (#6) | 74.1 (#6) | 74.1 (#6) | 74.1 (#6) |
| LLaDA | 72.6 (#7) | 72.6 (#7) | 72.6 (#7) | 72.6 (#7) | 72.6 (#7) | 72.6 (#7) |
| Fast-dLLM-LLaDA | 148.8 (#5) | 171.4 (#5) | 186.1 (#5) | 205.8 (#5) | 224.2 (#5) | 232.7 (#5) |
| D2F-LLaDA | 160.7 (#4) | 181.5 (#4) | 193.2 (#4) | 209.7 (#4) | 227.1 (#4) | 235.2 (#4) |
| SDTT-LLaDA | 234.4 (#3) | 263.9 (#3) | 283.4 (#3) | 309.7 (#3) | 334.6 (#3) | 346.1 (#3) |
| dParallel-LLaDA | 264.2 (#2) | 301.3 (#2) | 325.5 (#2) | 358.1 (#2) | 388.8 (#2) | 402.9 (#2) |
| **d3LLM-LLaDA** | **454.8** (#1) | **527.6** (#1) | **574.8** (#1) | **637.7** (#1) | **696.3** (#1) | **723.3** (#1) |

## A.6. Analysis of $y_{\min}$ in AUP

We further conduct a sensitivity analysis of the minimum accuracy threshold $y_{\min}$ in the AUP metric. We define $y_{\min} = y_1 - \Delta y$, where $y_1$ is the highest accuracy achieved. In the main experiments, we set $\Delta y = 5$. Here, we vary $\Delta y \in \{1, 2, 3, 5, 8, 10\}$ and report the AUP scores and rankings of all LLaDA-based methods on GSM8K-CoT in Table 13. As shown in the table, although the absolute AUP scores change with $\Delta y$, the relative ranking of all methods remains entirely consistent across all settings. This demonstrates that the AUP metric is robust to the choice of $y_{\min}$, and that the specific value does not affect the comparative evaluation. We adopt $\Delta y = 5$ as the default, which provides a reasonable tolerance for accuracy variation while filtering out impractical operating points.

**Table 14.** Comparison of our d3LLM framework with SOTA speculative decoding method, EAGLE-3 (Li et al., 2025c). We report the Tokens Per Forward (TPF), Accuracy, and Accuracy Under Parallelism (AUP) scores.

| Benchmark | Method | TPF ↑ | Accuracy ↑ | AUP Score ↑ |
|---|---|---|---|---|
| **GSM8K-CoT** | d3LLM-Dream | 4.94 | 81.4 | 391.3 |
| | d3LLM-LLaDA | 9.11 | 73.1 | 637.7 |
| | EAGLE-3 | 5.12 | 76.6 | 319.0 |
| **MATH** | d3LLM-Dream | 3.92 | 38.2 | 97.5 |
| | d3LLM-LLaDA | 5.74 | 30.4 | 107.6 |
| | EAGLE-3 | 5.72 | 39.8 | 142.1 |
| **MBPP** | d3LLM-Dream | 2.96 | 55.6 | 141.4 |
| | d3LLM-LLaDA | 4.21 | 40.6 | 88.4 |
| | EAGLE-3 | 5.69 | 60.2 | 298.6 |
| **HumanEval** | d3LLM-Dream | 3.20 | 57.1 | 129.5 |
| | d3LLM-LLaDA | 5.95 | 39.6 | 96.6 |
| | EAGLE-3 | 5.98 | 67.6 | 344.8 |
| **Long-GSM8K** | d3LLM-Dream | 4.80 | 77.2 | 348.6 |
| | d3LLM-LLaDA | 6.95 | 74.2 | 441.1 |
| | EAGLE-3 | 5.57 | 80.5 | 422.2 |

### A.7. Compare with Speculative Decoding Method

We further evaluate a state-of-the-art speculative decoding method, EAGLE-3 (with LLaMA-3.1-8B-Instruct) (Li et al., 2025c), on the same five datasets. The results are shown in Table 14. Notably, EAGLE-3 attains the highest overall AUP score. This is expected, as speculative decoding includes an additional verification step and therefore does not suffer from accuracy degradation under strong parallelism, unlike dLLMs. Moreover, our evaluation does not constrain total FLOPs, and speculative decoding methods may require more FLOPs than diffusion-based approaches. Nevertheless, our d3LLM framework substantially narrows the gap between diffusion-based models and state-of-the-art speculative decoding methods, offering valuable insights for future research directions.

### A.8. Combined with SGLang Inference Engine

All experiments in previous sections use the HuggingFace Transformers inference backend, which provides a convenient but not fully optimized environment for dLLM inference. We further integrate our d3LLM models into the SGLang inference engine (Zheng et al., 2024), a high-performance serving framework, with three key system-level optimizations: (i) we add *bidirectional attention* support in SGLang, enabling the engine to handle the non-causal attention pattern required by dLLMs; (ii) we implement our *multi-block decoding* algorithm directly within the SGLang runtime, allowing efficient parallel decoding across multiple blocks without framework-level overhead; and (iii) we apply *operator fusion* and other kernel-level optimizations tailored to the dLLM decoding pattern to reduce memory access and kernel launch overhead.

We evaluate *d3LLM-LLaDA* (8B dense) and *d3LLM-Dream* (7B dense) on GSM8K-CoT (zero-shot) using the SGLang engine with TP=1, across NVIDIA B200, H100, and A100 GPUs. Compared to the HuggingFace-based results reported in Section 4, the SGLang engine yields substantial throughput improvements across all hardware platforms, with the speedup scaling favorably on more powerful GPUs: as demonstrated in Table 15, our *d3LLM-LLaDA* achieves up to 1310 TPS on B200, 551 TPS on H100, and 251 TPS on A100, corresponding to approximately $4.8\times$, $5.1\times$, and $2.6\times$ speedup over the AR baseline (Qwen-2.5-7B-Instruct), while maintaining comparable accuracy.

### A.9. Details of Contenders

In our experiments, we compare our *d3LLM framework* with the following baselines and state-of-the-art dLLM methods:

- *Vanilla LLaDA* (Nie et al., 2025) is an open-source foundation dLLM trained from scratch that utilizes a vanilla Transformer to predict masked tokens.

- *Vanilla Dream* (Ye et al., 2025) is another popular foundation dLLM initialized from pre-trained autoregressive weights and employs context-adaptive noise rescheduling to enhance training efficiency and planning abilities.

- *Fast-dLLM* (Wu et al., 2025b) proposes a training-free acceleration framework that incorporates a block-wise approximate

**Table 15.** Throughput comparison using the SGLang (Zheng et al., 2024) inference engine on GSM8K-CoT (zero-shot). We report tokens per second (TPS) on B200, H100, and A100 GPUs, tokens per forward (TPF), and accuracy. Speedup ratios relative to the AR baseline (Qwen-2.5-7B-Instruct) on each GPU are shown in parentheses.

| Model | Threshold | Batch Size | B200 TPS ↑ | H100 TPS ↑ | A100 TPS ↑ | TPF ↑ | Acc (%) ↑ |
|---|---|---|---|---|---|---|---|
| Qwen-2.5-7B-Instruct | / | 1 | 274.7 (1.0×) | 108.6 (1.0×) | 96.8 (1.0×) | 1.00 | 74.1 |
| Qwen3-8B | / | 1 | 234.2 (0.9×) | 98.3 (0.9×) | 90.0 (0.9×) | 1.00 | 93.6 |
| **d3LLM-LLaDA** (8B dense) | 0.5 | 1 | 1241.0 (4.5×) | 545.3 (5.0×) | 251.6 (2.6×) | 9.91 | 75.4 |
| | 0.5 | 4 | 1310.2 (4.8×) | 551.9 (5.1×) | 250.0 (2.6×) | 8.56 | 75.1 |
| **d3LLM-Dream** (7B dense) | 0.4 | 1 | 586.8 (2.1×) | 280.5 (2.6×) | 125.6 (1.3×) | 4.89 | 80.9 |
| | 0.4 | 4 | 676.8 (2.5×) | 281.8 (2.6×) | 127.9 (1.3×) | 4.22 | 80.8 |

KV cache and a confidence-aware parallel decoding strategy to improve inference throughput.

- *Fast-dLLM-v2* (Wu et al., 2025a) adapts AR models into block diffusion models with minimal fine-tuning, utilizing hierarchical caching to achieve inference speeds surpassing standard AR decoding.

- *dParallel* (Chen et al., 2025) introduces a certainty-forcing distillation strategy that encourages the model to reach high predictive confidence rapidly, thereby enabling highly parallel decoding with fewer steps.

- *D2F* (Wang et al., 2025a) refurbishes diffusion models into an AR-diffusion hybrid via discrete diffusion forcing and asymmetric distillation, enabling exact KV cache utilization and inter-block parallel decoding.

- *SDTT* (Deschenaux & Gulcehre, 2025) introduces a self-distillation method for discrete diffusion language models that reduces the number of inference steps.

We use the officially released model weights of the above methods and incorporate them into our evaluation framework using their default settings and hyperparameters.

## A.10. Details of Datasets

We evaluate our *d3LLM framework* on the following five widely-used benchmark datasets:

- *GSM8K-CoT* (Gao et al., 2024): This dataset consists of high-quality grade school math problems requiring multi-step reasoning. We employ the Chain-of-Thought (CoT) evaluation setting, where the model is prompted to generate intermediate reasoning steps before producing the final answer.

- *HumanEval* (Chen et al., 2021): A code generation benchmark comprising 164 handwritten Python programming problems. Each problem includes a function signature, docstring, and unit tests to assess the functional correctness.

- *MBPP* (Austin et al., 2021b): A dataset containing around 1,000 crowd-sourced Python programming tasks designed for entry-level programmers. It covers programming fundamentals and task descriptions with automated test cases.

- *MATH* (Lewkowycz et al., 2022): A comprehensive collection of challenging mathematics problems derived from high school competitions. It spans diverse subjects such as algebra and geometry, serving to evaluate advanced quantitative reasoning capabilities.

- *Long-GSM8K* (Cobbe et al., 2021) A dataset consisting of 8.5K linguistically diverse grade school math word problems. It requires models to perform multi-step reasoning using elementary arithmetic operations to derive the correct solution. We use a setting with 5-shot few-shot prompt to evaluate under longer context windows (prompt length ≈ 1000).

## A.11. More Implementation Details

**Training Settings.** Our d3LLM begins with a block diffusion model (can be either LLaDA or Dream) with a block size of 32 as the teacher model. For fair comparison, we adopt the same distillation dataset as dParallel (Chen et al., 2025), which includes approximately 122k samples (about 65M tokens) for Dream and 92k samples (about 40M tokens) for LLaDA, sourced from the PRM12K (Lightman et al., 2024), AceCode (Zeng et al., 2025), GSM8K (training split) (Cobbe et al., 2021), and Numina-Math (Li et al., 2024a) datasets. The learning rate is set to $2 \times 10^{-5}$. We train 6 epochs for *d3LLM-LLaDA* and 3 for *d3LLM-Dream*.

We train our d3LLM models using LoRA (Hu et al., 2022) with rank $r = 256$ and $\alpha = 256$, targeting all linear layers (i.e., `q_proj`, `k_proj`, `v_proj`, `o_proj`, `gate_proj`, `up_proj`, `down_proj`). The training uses AdamW optimizer (Loshchilov & Hutter, 2019) with a learning rate of $2 \times 10^{-5}$ and weight decay of 0.01. For *d3LLM-LLaDA*, we train for 6 epochs with a constant learning rate scheduler and maximum sequence length of 384 tokens. For *d3LLM-Dream*, we train for 3 epochs with a cosine learning rate scheduler with 5% warmup ratio and maximum sequence length of 512 tokens. Both models use a batch size of 16 with gradient accumulation steps of 4, resulting in an effective batch size of 64. We employ DeepSpeed (Rasley et al., 2020) ZeRO-2 with CPU optimizer offloading for memory efficiency. Following dParallel (Chen et al., 2025), we adopt the certainty-forcing loss with entropy regularization (temperature=0.5, entropy weight 2.0 for LLaDA and 1.0 for Dream) to encourage confident predictions on correctly predicted tokens. We also use complementary masking loss to improve token utilization during training. The block size progressively increases from 16 to 32 across epochs, and the mask ratio linearly increases from 0.0 to 0.8 throughout training. All training is conducted on NVIDIA H100 GPUs with bfloat16 precision.

**Extracting Pseudo Trajectory of the Teacher dLLM.** As mentioned in Section 3, we extract the pseudo-trajectory of the teacher dLLM by using the teacher dLLM to generate and record its own decoding trajectory, where we constrain the teacher model to unmask exactly one token at each decoding step, and we continue generation beyond the EOS token so that the output length is exactly $n$. Since the teacher and student share the same model architecture, the teacher's trajectory reflects the model's own decoding preferences and thus provides a reasonable generation order for the student to learn; our ablation study in Table 5 further confirms that pseudo-trajectory distillation substantially outperforms random masking. One may observe that for a decoding window $w = \{s, \ldots, s + k\}$, directly using the global trajectory $\mathcal{T}_{s + \lceil kt \rceil}$ to construct the noisy sequence does not guarantee that the mask ratio within the window equals exactly $t$. An alternative approach might be to first sort the tokens within $w$ according to their generation order in the teacher's trajectory, and then mask the last $\lceil kt \rceil$ tokens to ensure a precise local mask ratio of $t$. However, we argue that this window-specific adaptation is inappropriate: our target is to train a dLLM with a block size of 32, and the teacher's pseudo-trajectory reflects the generation order under this full block size. Reordering or re-indexing tokens within smaller windows (i.e., $k < 32$) would disrupt the global ordering learned by the teacher, potentially introducing inconsistencies between the distillation objective and the actual inference-time decoding behavior. Therefore, we adopt the global trajectory without window-specific modifications, which preserves the teacher's original generation order.

**Inference Settings.** During inference, we set the maximum generation length to 256 tokens for most tasks and 512 tokens for Dream-Coder experiments. We use greedy decoding with temperature set to 0.0 or 0.1, depending on the specific task. The block size is fixed at 32 tokens for all experiments. For our d3LLM framework with multi-block generation, the entropy threshold ranges from 0.4 to 0.5, the block-add threshold is set to 0.1, and the decoded token threshold is 0.95. The cache delay iteration is typically set to 1–2, depending on the task requirements. For more details on hyperparameter configurations, please refer to our code repository.

### A.12. Further Improvements of d3LLM

In this work, we focus on building the d3LLM framework with algorithmic improvements in both training and inference. However, there remain several directions that could further improve the performance and efficiency of d3LLM. We discuss these potential extensions below.

**Combining with Speculative Decoding.** Speculative decoding (Leviathan et al., 2023; Chen et al., 2023; Qian et al., 2026) is a powerful framework for accelerating LLM inference by using draft models or self-speculation to predict multiple tokens at once, followed by a verification step to ensure generation quality. While speculative decoding has been primarily developed for AR models, recent work has explored its application to dLLMs (Gao et al., 2025; Xu et al., 2025; Chen et al., 2026). Our d3LLM framework could potentially be combined with speculative decoding techniques: for example, one could use a smaller dLLM as a draft model to propose candidate tokens, which are then verified by the larger model. This combination may further improve parallelism while preserving accuracy.

**Applying to Stronger Foundation dLLMs.** Most recently, there has been rapid progress in converting AR models to dLLMs (Wu et al., 2025a; Cheng et al., 2025; Li et al., 2025a; Fu et al., 2025; Bie et al., 2025). These new methods achieve notably better performance than the earlier foundation dLLMs (LLaDA and Dream) that we use in this work. We claim that, since our d3LLM is a *post-training* framework that consists of a distillation recipe and a decoding strategy that are largely

model-agnostic, it can serve as a plug-and-play component for these stronger dLLMs. Applying our pseudo-trajectory distillation and multi-block decoding strategy to advanced models such as ReFusion (Li et al., 2025a) or LLaDA 2.0 (Bie et al., 2025) may yield further improvements in both accuracy and parallelism. Alternatively, incorporating reinforcement learning techniques (Zhu et al., 2025; Li et al., 2025b; Wang et al., 2025b), continuous learning techniques (Qian et al., 2025), and quantization techniques (Lin et al., 2025) may further enhance the effectiveness of d3LLM.

In summary, this work focuses primarily on the algorithmic design of *distillation and decoding recipes* for dLLMs, achieving an ultra-fast and high-performance d3LLM framework. We have also demonstrated that system-level optimizations via the SGLang engine (Appendix A.8) can substantially amplify these gains. Many opportunities remain to further improve the performance and efficiency of diffusion language models, and we leave these potential extensions for future work.

## B. Related Work

In this part, we discuss the related topics.

**Diffusion Language Models.**

Diffusion language models have emerged as a promising alternative to AR, of which the key technique is masked diffusion (Austin et al., 2021a; Lou et al., 2023; Shi et al., 2024). Early efforts such as MDLM (Sahoo et al., 2024) and RADD (Ou et al., 2025) established effective training objectives for masked diffusion, substantially narrowing the performance gap with AR models. To further reduce the number of inference steps, distillation-based approaches have been explored: SDTT (Deschenaux & Gulcehre, 2025) distills multi-step teacher predictions into fewer student steps, and Duo (Sahoo et al., 2025) adapts consistency distillation to the discrete setting with a curriculum learning strategy, enabling competitive few-step generation.

Recently, a growing number of open-source foundation dLLMs have been developed, among which two notable examples are LLaDA (Nie et al., 2025), a native 8B dLLM trained from scratch, and Dream (Ye et al., 2025), a 7B dLLM initialized from an autoregressive LLM. Most recently, LLaDA 2.0 (Bie et al., 2025) scaled up the open-source dLLM to 100B total parameters through conversion from AR models, delivering superior performance at the frontier scale.

With the growing interest of the research community, an increasing number of works have been proposed to improve dLLMs in terms of efficiency or performance. On one hand, acceleration of dLLMs has been an active research area in recent years (Wu et al., 2025b; Ma et al., 2025a; Gao et al., 2025; Ma et al., 2025b; Yang et al., 2025a; Wu & Zhang, 2025; Lin et al., 2026). Fast-dLLM (Wu et al., 2025b) presents a training-free acceleration method using a block-wise approximate KV-cache mechanism tailored for dLLMs. dKV (Ma et al., 2025a) proposes a delayed KV-cache mechanism that caches key and value states with a delayed and conditioned strategy to accelerate inference. dInfer (Ma et al., 2025b) develops iteration smoothing, hierarchical and credit decoding, and refresh strategies alongside system-level optimizations to accelerate dLLMs across multiple dimensions. Most recently, dParallel (Chen et al., 2025) introduces a learnable parallel decoding mechanism with certainty-forcing loss for distillation (Hinton et al., 2015; Zhou & Jiang, 2004), achieving significant speedup. ParallelBench (Kang et al., 2026) provides the first benchmark specifically designed for dLLMs, revealing that parallel decoding can suffer dramatic quality degradation and highlighting the pressing need for methods that overcome the speed–quality trade-off.

Another line of work focuses on improving the performance of dLLMs. MMaDA (Yang et al., 2025b) employs a unified policy-gradient-based reinforcement learning algorithm to enhance dLLM performance across multiple modalities. ReFusion (Li et al., 2025a) initializes dLLMs from Qwen-3-8B and adopts a slot-level parallel decoding mechanism, achieving 34% performance gains over prior masked diffusion models. TraDo (Wang et al., 2025c) proposes a trajectory-aware reinforcement learning framework that incorporates preferred inference trajectories into post-training, achieving strong reasoning performance on math and coding tasks.

**Speculative Decoding.**  A separate line of work seeks to improve the efficiency of AR models through speculative decoding (Leviathan et al., 2023; Chen et al., 2023; Cai et al., 2024; Ankner et al., 2024). These frameworks typically leverage models of different sizes (a large model together with one or more smaller models) to accelerate inference. EAGLE (Li et al., 2024b) employs feature-level autoregression with token-conditioned drafting to enable efficient speculative sampling. EAGLE-3 (Li et al., 2025c) further leverages multi-layer feature fusion to improve scalability. OSD (Liu et al., 2024) adapts drafts in an online manner, continuously improving token acceptance and reducing latency. This approach

could be further enhanced by incorporating modern online learning techniques (Zhao et al., 2024; Qian et al., 2024).

In addition, Medusa (Cai et al., 2024) augments LLM inference by adding extra heads to predict multiple subsequent tokens at once rather than one token at a time. Hydra (Ankner et al., 2024) extends Medusa by introducing sequentially-dependent draft heads, where each head considers previously speculated tokens when predicting the next one. CLLM (Kou et al., 2024) enables parallel decoding in AR models by training the model to consistently predict the fixed point given any state on a Jacobi trajectory. REST (He et al., 2024), Lookahead decoding (Fu et al., 2024), and PLD+ (Somasundaram et al., 2025) explore an alternative approach: rather than relying on a draft model, they obtain speculative candidates directly from context or future tokens. A key feature of speculative decoding is that it achieves high throughput while preserving generation quality, as it includes a verification step by the target model to ensure the generated tokens are correct.

## C. Limitation

Our d3LLM framework is evaluated primarily on LLaDA (Nie et al., 2025) and Dream (Ye et al., 2025), which are the two most widely studied open-source foundation dLLMs with extensive prior work available for comparison. Although more recent and potentially stronger dLLMs have emerged (e.g., SDAR (Cheng et al., 2025), ReFusion (Li et al., 2025a), LLaDA 2.0 (Bie et al., 2025)), we mention our approach, comprising a distillation recipe and an inference strategy, is model-agnostic and can serve as a plug-and-play component for these newer models to achieve improvements on accuracy and parallelism.

