# OpenReview forum: "d3LLM: Ultra-Fast Diffusion LLM using Pseudo-Trajectory Distillation"
_ICML.cc/2026/Conference — ICML 2026 regular_

### Official Review · Reviewer_Z7Lq · 2026-02-18

**Soundness:** 3
**Presentation:** 3
**Significance:** 2
**Originality:** 3
**Overall Recommendation:** 4
**Confidence:** 3

**Summary:**

D3LLM addresses the fundamental accuracy-parallelism trade-off in Diffusion Large Language Models (dLLMs) through three key componenets:
* Trajectory-Aware Distillation: A novel forward process that replaces uniform masking with a scheme informed by teacher dLLM trajectories.
* An entropy-based decoding strategy that transcends single-block constraints, integrated with a KV-cache refreshing mechanism.
* Introduction of the AUP metric to rigorously quantify the frontier.

**Compliance With Llm Reviewing Policy:**

Affirmed.

**Final Justification:**

The authors addressed my concerns.

**Key Questions For Authors:**

* Could the authors provide a more detailed motivation for the specific choice of the weight function $W$ used in the AUP metric? Furthermore, an ablation study or sensitivity analysis regarding different functional forms of $W$ would be valuable to demonstrate that the reported improvements are robust and not overly dependent on this specific parameterization.
* Table 3 reports a 5x increase in TPS for D3LLM relative to Qwen-7B, a gain that appears to be primarily driven by the TPF factor (4.94x). However, D3LLM introduces additional computational overhead via a bidirectional attention scheme and a KV-cache refreshing mechanism, both of which typically increase the latency of a single forward pass. Could the authors provide a details to settle this gap?

**Limitations:**

yes

**Strengths And Weaknesses:**

Strengths -
* The paper is well-structured and clearly written, with the underlying motivations and methodologies presented in a logically sound and accessible manner.
* The proposed pseudo-trajectory-based distillation represents a significant departure from standard methods by introducing a more principled framework for constructing the generative forward process.
* The comprehensive empirical evaluation provides compelling evidence that successfully validates the core claims of the proposed method.

Weaknesses -
* While the introduction and section 2 provide a motivation for the AUP metric, the submission would be significantly strengthened by empirical evidence demonstrating the failure of the standard AUC metric in this specific context. Can the authors provide a case study or experimental results where AUC fails to capture the accuracy-parallelism trade-off that AUP successfully identifies?
* Several technical specifications essential for reproducibility and a full assessment of the model's efficiency are currently missing from the main text. What kind attention pattern does the model uses during training/inference (fully bidirectional or mixture of causal and bidirectional as commonly used in block dLLM)? During inference, how often do the KV cache gets refreshed?
* While increasing parallelism is a valuable objective, its primary utility lies in enhancing overall system efficiency. To provide a comprehensive evaluation of d3LLM’s efficiency, the authors should report the total number of generated tokens alongside existing metrics. Coupling this with TPF and TPS would establish a more transparent basis for comparing computational efficiency against baseline models.

---

> ### Author Rebuttal · Authors · 2026-03-29
>
> We sincerely appreciate the reviewer's feedback. In the following, we address each of your concerns.
>
> ---
>
> **Q1.** "About the choice of the weight function W used in the AUP metric."
>
> **A1.** We thank the reviewer for this question. Our motivation for choosing the exponential penalty function is that it imposes a strong penalty on accuracy degradation, as the exponential form naturally amplifies the cost of performance drops in a principled manner. This design ensures that methods achieving high parallelism at the expense of accuracy degradation are appropriately penalized.
>
> We further conduct an analysis that evaluates different penalty functions, including exponential (Exp), power (Pow), and linear (Lin) forms:
>
> - **Exponential**: $W(y) = \min(\exp(-\alpha \cdot (1 - y/y_{\max})), 1)$ with $\alpha \in \\{1, 3, 5\\}$
> - **Power**: $W(y) = (y/y_{\max})^\alpha$ with $\alpha \in \\{1, 3, 5\\}$
> - **Linear**: $W(y) = \max(1 - \alpha \cdot (1 - y/y_{\max}), 0)$ with $\alpha \in \\{1, 5, 10\\}$
>
> The results for LLaDA-based models are shown below (AUP scores on GSM8K-CoT dataset):
>
> ||Exp(α=1)|Exp(α=3)|Exp(α=5)|Pow(α=1)|Pow(α=3)|Pow(α=5)|Lin(α=1)|Lin(α=5)|Lin(α=10)|
> |-|:-:|:-:|:-:|:-:|:-:|:-:|:-:|:-:|:-:|
> |LLaDA|72.6 (#5)|72.6 (#5)|72.6 (#5)|72.6 (#5)|72.6 (#5)|72.6 (#5)|72.6 (#5)|72.6 (#5)|72.6 (#5)|
> |Fast-dLLM-LLaDA|206.6 (#4)|205.8 (#4)|204.9 (#4)|206.6 (#4)|205.8 (#4)|204.9 (#4)|206.6 (#4)|204.9 (#3)|202.7 (#3)|
> |D2F-LLaDA|210.8 (#3)|209.7 (#3)|208.7 (#3)|210.8 (#3)|209.8 (#3)|208.7 (#3)|210.8 (#3)|208.7 (#4)|205.4 (#4)|
> |dParallel-LLaDA|370.9 (#2)|358.1 (#2)|346.0 (#2)|370.8 (#2)|357.9 (#2)|345.6 (#2)|370.8 (#2)|344.0 (#2)|310.4 (#2)|
> |d3LLM-LLaDA|659.4 (#1)|637.7 (#1)|616.8 (#1)|659.2 (#1)|637.3 (#1)|616.3 (#1)|659.2 (#1)|614.0 (#1)|557.4 (#1)|
>
> As shown in the table, the ranking of methods remains consistent across all penalty functions. We will include this analysis in the revised version.
>
> ---
>
> **Q2.** "Could the authors provide details to settle this gap (between speedup and computational overhead of dLLM)?"
>
> **A2.** We thank the reviewer for pointing this out. We would like to clarify a potential misunderstanding: the 5× TPS speedup reported in Table 3 is achieved by d3LLM-LLaDA (TPF=9.11), not by d3LLM-Dream (TPF=4.94). While dLLM-based methods benefit from higher TPF, they do incur additional overhead due to bidirectional attention and KV-cache management. However, when TPF is sufficiently high, as in our d3LLM, the parallelism gains substantially outweigh the computational overhead, therefore achieving significant end-to-end speedup.
>
> We provide a comparison of TPF, TPS, average number of generated tokens, and accuracy on GSM8K-CoT:
>
> ||TPF|H100 TPS|A100 TPS|#Avg Tokens|Acc (%)|
> |-|:-:|:-:|:-:|:-:|:-:|
> |Qwen-2.5-7B-it|1.0|57.3 (1.0×)|50.4 (1.0×)|201.7|74.1|
> |LLaDA|1.00|27.9 (0.5×)|19.2 (0.4×)|228.5|72.6|
> |Fast-dLLM-LLaDA|2.77|114.3 (2.0×)|79.1 (1.6×)|220.7|74.7|
> |D2F-LLaDA|2.88|102.1 (1.8×)|76.2 (1.5×)|190.1|73.2|
> |dParallel-LLaDA|5.14|172.2 (3.0×)|105.9 (2.1×)|193.6|72.6|
> |**d3LLM-LLaDA**|**9.11**|**288.9 (5.0×)**|**183.3 (3.6×)**|215.5|73.1|
>
> The results demonstrate that d3LLM achieves the highest TPF, which enjoys substantial wall-clock speedups even with the bidirectional attention overhead. We will include this analysis in the revised version.
>
> ---
>
>
> **Q3.** "Can the authors provide a case study or experimental results where AUC fails..."
>
> **A3.** We thank the reviewer for this question. As a concrete example in Section 4.2 (line 241-251), the standard AUC metric fails to capture the accuracy-parallelism trade-off on the MBPP dataset with LLaDA-based models. Specifically, although many methods achieve high parallelism, their accuracy degrades substantially compared with the best-performing model Qwen-2.5-7B. The standard AUC metric assigns high scores to these methods (e.g., AUC > 90 for dParallel-LLaDA) due to their high TPF, failing to penalize the significant accuracy loss. In contrast, AUP appropriately penalizes methods that sacrifice accuracy for parallelism, thus faithfully capturing the accuracy-parallelism trade-off. We will emphasize this case study in the revised manuscript.
>
> ---
>
> **Q4.** Other implementation and presentation details.
>
> **A4.** Thank you for your constructive feedback. (i) d3LLM employs a *fully bidirectional attention* mechanism during both training and inference, consistent with the standard dLLMs as in LLaDA and Dream. During inference, the KV-cache is refreshed every 3 decoding rounds or when a block is completed. (ii) We have included detailed token count statistics in the revised version to provide a more comprehensive efficiency analysis. We appreciate your valuable suggestions for improving the clarity of our paper.
>
> ---
>
> **In summary,** we have included additional experiments following your suggestions, and will revise the paper accordingly. If our responses have addressed your concerns, we would be grateful if you could consider raising the score. Thank you!

---

> > ### Author Rebuttal · Reviewer_Z7Lq · 2026-04-03
> >
> > I thank the authors for their detailed response. It provided the missing information required for me to complete my assessment, and therefore, I will maintain my positive score.

---

### Official Review · Reviewer_vrQa · 2026-02-20

**Soundness:** 3
**Presentation:** 3
**Significance:** 3
**Originality:** 3
**Overall Recommendation:** 5
**Confidence:** 3

**Summary:**

This paper addresses a fundamental challenge in diffusion-based large language models (dLLMs): the trade-off between decoding parallelism and accuracy. While dLLMs naturally support parallel token generation, increasing parallelism often leads to premature token commitments and degraded performance. The authors aim to push the accuracy–parallelism frontier upward, improving throughput without sacrificing quality.

The proposed framework, d3LLM, consists of three main components:

Pseudo-Trajectory Distillation: During training, the student model learns from a teacher model's decoding trajectory (i.e., the order in which tokens are resolved). This supervision encourages the student to identify which tokens can be safely decoded earlier, improving robustness under higher parallelism.

Entropy-based Multi-Block Decoding with KV-Refresh: During inference, tokens with low entropy (high confidence) are decoded across multiple blocks in parallel. To mitigate error accumulation caused by incomplete context in multi-block decoding, periodic KV-cache refresh is introduced.

Accuracy Under Parallelism (AUP): A new evaluation metric that measures the area under the accuracy–parallelism curve, with penalties for significant accuracy drops, providing a unified assessment of quality–speed trade-offs.

Experiments across multiple open-source diffusion LLMs (e.g., LLaDA, Dream, Dream-Coder) demonstrate substantial throughput improvements while maintaining competitive accuracy. The method consistently improves AUP compared to vanilla diffusion LLMs.

**Compliance With Llm Reviewing Policy:**

Affirmed.

**Final Justification:**

The paper proposes D3LLM to address the accuracy-parallelism trade-off in diffusion LLMs. I found the approach technically sound, with a clear presentation and reasonably strong empirical results.

In my initial review, I raised concerns about the motivation and robustness of the AUP metric, missing implementation details, and the completeness of the efficiency analysis. The rebuttal addresses these points satisfactorily: the additional analysis on different weighting functions supports the robustness of AUP, the clarification of attention design and KV-cache refreshing improves reproducibility, and the added efficiency statistics (including token counts) help justify the reported speedups.

Overall, the rebuttal resolves my main concerns and increases my confidence in the work, and I have updated my score accordingly.

**Key Questions For Authors:**

How sensitive is AUP to the choice of weighting and penalty functions? Would alternative reasonable formulations change the ranking of methods?

How robust is pseudo-trajectory distillation across different teacher architectures or weaker teacher models?

What is the precise computational overhead introduced by KV-refresh, and how does refresh frequency affect the speed–accuracy trade-off?

Does the method generalize to significantly longer sequence lengths or different task domains?

Under equal wall-clock or hardware budgets, how does d3LLM compare to other acceleration strategies?

**Limitations:**

The paper discusses computational trade-offs and teacher dependency, which is appreciated. However, a more explicit analysis of cost overhead and practical deployment constraints would strengthen the limitations section.

**Strengths And Weaknesses:**

Soundness

Strengths

The problem formulation is clear and well-motivated.

The combination of trajectory-based distillation and entropy-driven inference is logically consistent and addresses both training and inference limitations.

The proposed KV-refresh mechanism is a practical solution to context drift in multi-block decoding.

Experimental results support the claim that the method improves the accuracy–parallelism trade-off.

Weaknesses

The fairness and robustness of the proposed AUP metric may require further validation. It is unclear how sensitive conclusions are to the specific weighting or penalty design.

The effectiveness of pseudo-trajectory distillation may depend on the quality or architecture of the teacher model.

KV-refresh introduces additional computational overhead, and the cost–benefit trade-off should be quantified more explicitly.

Presentation

Strengths

The paper is clearly structured and follows a coherent narrative from problem definition to method design and evaluation.

The description of training and inference components is sufficiently detailed for reproducibility.

Weaknesses

The differentiation from closely related acceleration and parallel decoding methods could be further sharpened.

Additional intuitive explanations or case studies for AUP would improve clarity.

Significance

The work addresses a practically important problem in diffusion LLM deployment. Improving decoding parallelism without sacrificing accuracy is crucial for real-world adoption. The trajectory distillation idea may generalize to other parallel generation paradigms. However, the broader impact depends on reproducibility and system-level cost analysis.

Originality

The pseudo-trajectory distillation mechanism is a notable contribution, as it supervises decoding order rather than only final outputs. The integration of training-time supervision and inference-time parallel decoding optimization is a meaningful system-level innovation. However, individual components (multi-block decoding, entropy selection, cache refresh) build upon existing ideas, and the novelty lies primarily in their integration.

---

> ### Author Rebuttal · Authors · 2026-03-29
>
> We sincerely thank the reviewer for the helpful comments. We answer your technical questions below.
>
> ---
>
> **Q1.** "How sensitive is AUP to the choice of weighting and penalty functions?"
>
> **A1.** Thank you for this insightful question. We conducted additional experiments to evaluate the sensitivity of AUP to different penalty functions. The detailed experimental results are provided in https://anonymous.4open.science/r/d3LLM/rebuttal/penalty_analysis.md. The results demonstrate that the ranking of methods remains consistent across all nine penalty function configurations, indicating that AUP is robust to the choice of penalty functions and provides reliable method comparisons. We will include this in the revised version.
>
> ---
>
> **Q2.** "How robust is pseudo-trajectory distillation across different teacher architectures or weaker teacher models?"
>
> **A2.** Thank you for this question. Regarding different teacher architectures, we have validated our approach on three different foundation models: LLaDA, Dream, and Dream-Coder. Our method consistently achieves the highest AUP scores across all three architectures, demonstrating the generalizability of our d3LLM framework.
>
> Regarding weaker teacher models, we conducted an ablation study in Table 5. The results show that when the teacher produces poor decoding orders (using random masking instead of pseudo-trajectory), the performance decreases significantly. This indicates that the quality of the teacher's trajectory does impact distillation performance.
>
> ---
>
> **Q3.** "What is the precise computational overhead introduced by KV-refresh, and how does refresh frequency affect the speed–accuracy trade-off?"
>
> **A3.** Thank you for raising this important question. The KV-refresh mechanism is triggered only when a block is completed or when the a block is just finished decoding, thus its computational overhead is small. To provide a precise analysis of how KV-refresh affects the speed–accuracy trade-off, we conducted additional experiments varying the refresh interval on the GSM8K-CoT dataset:
>
> |Refresh Interval|TPF|TPS (A100)|Accuracy (%)|
> |:-:|:-:|:-:|:-:|
> |1|9.13|176.43|73.16|
> |2|9.12|178.56|73.12|
> |3|9.11|183.31|73.09|
> |4|9.06|187.68|72.48|
> |5|8.95|186.39|70.36|
> |10|8.72|180.96|70.14|
>
> These results demonstrate that refresh frequency affects the speed–accuracy trade-off: excessively low refresh frequencies reduce computational cost but degrade KV-cache quality, leading to lower TPF and Acc, while excessively high refresh frequencies maintain high TPF and accuracy at the expense of increased computational overhead that reduces overall TPS. Our choice of the refresh interval as 3 achieves a balance between parallelism and accuracy. We will include this analysis in the revised manuscript.
>
> ---
>
> **Q4.** "Does the method generalize to significantly longer sequence lengths or different task domains?"
>
> **A4.** Thank you for raising this important question. We have evaluated d3LLM across diverse task domains, including mathematical reasoning, conversational generation, and coding tasks, as shown in Tables 1, 2, and 11. These results demonstrate that d3LLM is a general framework that maintains good performance across different domains. We plan to further investigate longer contexts and more complex tasks, such as multi-turn agentic tasks, in future work.
>
> ---
>
> **Q5.**  "Under equal wall-clock or hardware budgets, how does d3LLM compare to other acceleration strategies?"
>
> **A5.** Thank you for this question. To provide a fair comparison under equal computational budgets, we conducted additional experiments controlling parallelism (TPF) across different methods. Specifically, we evaluated all methods on the MBPP dataset and compared their accuracy:
>
> |TPF|Fast-dLLM-LLaDA|D2F-LLaDA|dParallel-LLaDA|d3LLM-LLaDA|
> |:-:|:-:|:-:|:-:|:-:|
> |1.00|41.58|39.10|41.62|**42.00**|
> |1.28|41.39|39.09|41.55|**41.99**|
> |1.56|40.84|39.08|41.34|**41.96**|
> |1.85|39.90|39.04|40.98|**41.90**|
> |2.13|38.60|39.00|40.48|**41.83**|
>
> These results demonstrate that under equal computational budgets (fixed TPF), d3LLM consistently achieves higher accuracy than contenders. We will include this analysis in the revised manuscript.
>
> ---
>
> **Q6.** Regarding the novelty... Compared with related methods.
>
> **A6.** We respectfully emphasize our core contribution: we identify the fundamental accuracy–parallelism trade-off in dLLMs, propose the AUP metric to measure it, and then design d3LLM framework to push this trade-off frontier. We believe that identifying and systematically addressing this fundamental trade-off represents a meaningful contribution to the field. Thank you again for your question.
>
> ---
>
> **In summary,** we have included additional experiments following your suggestions, including analysis of AUP's penalty functions, refresh frequency, and comparisons under equal computational budgets. If our responses have addressed your concerns, we would be grateful if you would consider raising the score. Thank you!

---

> > ### Author Rebuttal · Reviewer_vrQa · 2026-04-01
> >
> > The rebuttal provides substantial additional experimental evidence that addresses several of my key concerns. In particular, the sensitivity analysis of AUP under multiple penalty functions is convincing and supports the robustness of the metric. The analysis of KV-refresh frequency clearly quantifies the speed–accuracy trade-off, and the comparison under equal computational budgets further strengthens the empirical evaluation.
> >
> > The validation across multiple teacher architectures and the ablation with weaker teachers also help clarify the robustness of pseudo-trajectory distillation. Overall, these additions significantly improve the credibility of the method and its evaluation.
> >
> > Some aspects, such as generalization to longer sequences, remain less explored, but the main concerns have been adequately addressed. Therefore, I have a slightly more positive assessment after the rebuttal.

---

> > > ### Author Response · Authors · 2026-04-02
> > >
> > > We sincerely thank the reviewer for the constructive feedback and for raising the score, and we will carefully incorporate your valuable suggestions into the revised manuscript. Thank you!

---

### Official Review · Reviewer_oKYB · 2026-03-10

**Soundness:** 3
**Presentation:** 3
**Significance:** 3
**Originality:** 3
**Overall Recommendation:** 4
**Confidence:** 4

**Summary:**

This paper proposes d3LLM, a diffusion LLM acceleration framework that combines multiple techniques, including pseudo-trajectory distillation, curriculum noise, curriculum window, multi-block decoding, KV-refresh, and early stopping. These components are designed to better balance the trade-off between generation speed and model accuracy. Experimentally, compared with methods such as Fast-dLLM, D2F, and DParallel, the proposed approach achieves more substantial speedups while maintaining competitive accuracy. The paper also introduces AUP as a metric to jointly evaluate efficiency and performance, providing a more comprehensive assessment criterion.

**Compliance With Llm Reviewing Policy:**

Affirmed.

**Final Justification:**

My concerns have been partially addressed. However, it is still not entirely clear why the teacher model’s trajectories are beneficial for distillation. The rebuttal provides some empirical evidence, but since this is a central component of the paper, more analysis would be valuable, such as visualizing the discrepancy between teacher and student trajectories. Nevertheless, given the overall quality of the paper, I will maintain a positive score.

**Key Questions For Authors:**

Please see the weaknesses.

**Limitations:**

Limitations have been discussed in the appendix C.

**Strengths And Weaknesses:**

## Strengths
1. The accuracy–efficiency trade-off discussed in this paper is an important problem. The proposed method provides an in-depth exploration of this trade-off, including how to achieve the best possible performance while increasing parallelism, as well as how to properly evaluate such a balance. These investigations are valuable to the community.
2. The method is able to achieve a 3–5× speedup while delivering performance that is stronger than existing baselines.
3. This work presents a complete and practical acceleration framework, which could help facilitate the real-world deployment of diffusion LLMs.

## Weaknesses
1. The methodological contribution of this paper may be somewhat limited. While it combines multiple techniques, the novelty of each individual component appears relatively modest. For example, block-wise parallel decoding has already been explored in Fast-dLLM, and KV-refresh is essentially a standard application of KV-cache updating in the diffusion LLM setting. As a result, the overall level of novelty may be somewhat insufficient.
2. I also have some concerns about pseudo-trajectory distillation. This design seems to rely on the assumption that the trajectory produced by the teacher model represents a relatively good decoding order.

---

> ### Author Rebuttal · Authors · 2026-03-29
>
> Thanks for your constructive and helpful comments. Below, we address your concerns.
>
> ---
>
> **Q1.** "The methodological contribution of this paper may be somewhat limited."
>
> **A1.** Thank you for this question. First, we would like to clarify a key distinction: Fast-dLLM performs parallel decoding within a single block, while our method decodes across multiple blocks simultaneously. Besides, we respectfully emphasize our core contribution: we identify the fundamental accuracy–parallelism trade-off in dLLMs, propose the AUP metric to measure it, and then design d3LLM framework to push this trade-off frontier. We believe that identifying and systematically addressing this fundamental trade-off represents a meaningful contribution to the field. Thank you again for your comment.
>
> ---
>
> **Q2.** "I also have some concerns about pseudo-trajectory distillation. This design seems to rely on the assumption that the trajectory produced by the teacher model represents a relatively good decoding order."
>
> **A2.** Thank you for pointing this out. We clarify that in our setting, the teacher and student share the same model architecture, so the teacher's trajectory represents a reasonable decoding order for the student to learn. Since the trajectory reflects the model's own decoding preferences, it is natural for the student to follow this order. Our ablation study in Table 5 demonstrates that pseudo-trajectory distillation significantly outperforms random masking, improving TPF by 17.8% while maintaining similar accuracy. This validates that the teacher's trajectory provides useful guidance. We will demonstrate this more clearly in the revised version. Thank you again for your question.

---

> > ### Author Rebuttal · Reviewer_oKYB · 2026-04-01
> >
> > My concerns have been partially addressed. However, it is still not entirely clear why the teacher model’s trajectories are beneficial for distillation. The rebuttal provides some empirical evidence, but since this is a central component of the paper, more analysis would be valuable, such as visualizing the discrepancy between teacher and student trajectories. Nevertheless, given the overall quality of the paper, I will maintain a positive score.

---

> > > ### Author Response · Authors · 2026-04-02
> > >
> > > Thank you for your positive evaluation of our work! We would like to take this opportunity to briefly reiterate the motivation behind pseudo-trajectory distillation: the teacher's decoding trajectory provides intermediate supervision indicating which tokens can be safely decoded earlier with high confidence, thereby guiding the student to learn a more efficient unmasking order that improves parallelism.
> > >
> > > We will include additional visualizations in the revised manuscript to further illustrate the discrepancy between teacher and student trajectories. We sincerely appreciate your constructive suggestions. Thank you!

---

### Official Review · Reviewer_9z8T · 2026-03-18

**Soundness:** 2
**Presentation:** 3
**Significance:** 2
**Originality:** 2
**Overall Recommendation:** 4
**Confidence:** 4

**Summary:**

This paper takes an important step toward quantifying the speed–quality trade-off in dLLMs by introducing the AUP metric which computes the weighted area under the accuracy-parallelism curve. They also propose a distillation strategy to accelerate MDLMs and label their method d3llm. Their proposed method achieves significant speed-ups over the base dLLMs: Dream and Llada and is 5x faster on the gsm8k benchmark over the Qwen model.

**Compliance With Llm Reviewing Policy:**

Affirmed.

**Final Justification:**

The authors addressed my concerns.

**Key Questions For Authors:**

Please compare your method to SDTT [1]

[1] Deschenaux, Justin, and Caglar Gulcehre. "Beyond autoregression: Fast llms via self-distillation through time." ICLR 2025.

**Limitations:**

yes.

**Strengths And Weaknesses:**

Strengths:
1. This paper takes an important step toward quantifying the speed–quality trade-off in dLLMs.

2. The paper also proposes a distillation strategy to accelerate MDLMs.

Weaknesses:
1.	A major weakness is the lack of a comparison with SDTT [1], which should be included as a baseline. I would be happy to raise my score if the authors include this baseline in the rebuttal.

2.	The related work section is weak. The authors should compare and contrast their proposed method with prior diffusion distillation approaches in the literature, such as Discrete Consistency Distillation [2].

3.	Several important citations are missing, including MDLM [3] and RADD [4], which concurrently proposed the weighted cross-entropy loss used to train dLLMs.

4.	The AUP metric introduces two hyperparameters, $\alpha$ and $y_{\min}$. While $\alpha$ is thoroughly ablated, $y_{\min}$ is simply set to $y_1 - 5$ without any concrete justification for choosing 5. This weakens the motivation for the proposed metric.


Minor Comments:
1. The phrase “efficiency or performance …” in the abstract is unclear. I assume “efficiency” refers to faster inference, while “performance” refers to accuracy on downstream tasks.




[1] Deschenaux, Justin, and Caglar Gulcehre. "Beyond autoregression: Fast llms via self-distillation through time." ICLR 2025.
[2] Sahoo, Subham Sekhar, et al. "The diffusion duality." ICML 2025.
[3] Sahoo et al., "Simple and Effective Masked Diffusion Language Models", NeurIPS 2024.
[4] Ou et al., "Your Absorbing Discrete Diffusion Secretly Models the Conditional Distributions of Clean Data", ICLR 2025

---

> ### Author Rebuttal · Authors · 2026-03-29
>
> We sincerely appreciate the reviewer's feedback. We have added additional experiments and will revise the paper according to your suggestions. In the following, we address each of your concerns.
>
> ---
>
> **Q1 & Q2.** "Please compare your method to SDTT [1]... compare and contrast their proposed method with prior diffusion distillation approaches in the literature, such as Discrete Consistency Distillation [2]."
>
> **A1 & A2.** We appreciate the reviewer's suggestion. SDTT [1] is a distillation method for discrete diffusion models that reduces the number of inference steps by training the student to match the KL divergence of the teacher's multi-step predictions. DCD [2] constructs deterministic trajectories in discrete space via an underlying Gaussian diffusion process to accelerate diffusion models. We have conducted an additional experimental comparison with SDTT, and the results are presented below (Acc / TPF / AUP scores are listed):
>
> ||GSM8K-CoT|MATH|MBPP|HumanEval|Long-GSM8K|
> |-|:-:|:-:|:-:|:-:|:-:|
> |LLaDA|72.55 / 1.00 / 72.55|32.20 / 1.00 / 32.20|41.72 / 1.00 / 41.72|38.28 / 1.00 / 38.28|78.58 / 1.00 / 78.58|
> |SDTT-LLaDA|71.62 / 4.53 / 309.71|30.21 / 2.79 / 58.57|40.02 / 2.52 / 63.25|38.95 / 4.81 / 82.34|73.85 / 4.58 / 293.65|
> |d3LLM-LLaDA|73.09 / 9.11 / 637.65|30.36 / 5.74 / 107.64|40.60 / 4.21 / 88.36|39.63 / 5.95 / 96.64|74.22 / 6.95 / 441.13|
>
> As shown in the table, SDTT performs full-sequence distillation, which leads to some accuracy loss; in contrast, d3LLM leverages the teacher's decoding order and employs curriculum learning strategy, achieving a better accuracy–efficiency balance for dLLMs. We will incorporate these results along with a detailed discussion of SDTT and DCD in the revised version. We thank the reviewer again for this valuable suggestion.
>
> ---
>
> **Q3.** "Several important citations are missing, including MDLM [3] and RADD [4], which concurrently proposed the weighted cross-entropy loss used to train dLLMs."
>
> **A3.** Thank you for your comments. We acknowledge that our current related work section primarily focused on recent works following the emergence of large-scale dLLMs, and allocated less space to pioneering works on masked diffusion modeling, including MDLM [3] and RADD [4], which established key training objectives and foundational methods for masked diffusion. We will add these citations and give credit to these works in the revised version. Thank you again for your helpful suggestion!
>
> ---
>
> **Q4.** "The AUP metric introduces two hyperparameters, alpha and $y_{\min}$. While $\alpha$ is thoroughly ablated, $y_{\min}$ is simply set to $y_1 - 5$ without any concrete justification for choosing 5."
>
> **A4.**  We thank the reviewer for this question. We have conducted an additional sensitivity analysis of $y_{\min}$, and the results are presented below. We report the AUP score and ranking of each method on the GSM8K-CoT dataset under varying values of $\Delta y$ (where $y_{\min} = y_1 - \Delta y$):
>
> ||Δy=1|Δy=2|Δy=3|Δy=5|Δy=8|Δy=10|
> |-|:-:|:-:|:-:|:-:|:-:|:-:|
> |Qwen-2.5-7B-it|74.1 (#6)|74.1 (#6)|74.1 (#6)|74.1 (#6)|74.1 (#6)|74.1 (#6)|
> |LLaDA|72.6 (#7)|72.6 (#7)|72.6 (#7)|72.6 (#7)|72.6 (#7)|72.6 (#7)|
> |Fast-dLLM-LLaDA|148.8 (#5)|171.4 (#5)|186.1 (#5)|205.8 (#5)|224.2 (#5)|232.7 (#5)|
> |D2F-LLaDA|160.7 (#4)|181.5 (#4)|193.2 (#4)|209.7 (#4)|227.1 (#4)|235.2 (#4)|
> |dParallel-LLaDA|264.2 (#2)|301.3 (#2)|325.5 (#2)|358.1 (#2)|388.8 (#2)|402.9 (#2)|
> |SDTT-LLaDA|234.4 (#3)|263.9 (#3)|283.4 (#3)|309.7 (#3)|334.6 (#3)|346.1 (#3)|
> |d3LLM-LLaDA|454.8 (#1)|527.6 (#1)|574.8 (#1)|637.7 (#1)|696.3 (#1)|723.3 (#1)|
>
> As shown in the table, although the absolute AUP scores vary with Δy, the relative ranking of all methods remains consistent across all settings. This demonstrates that the AUP metric is robust to the choice of $y_{\min}$, and that the specific value does not affect the comparative evaluation. We will add these results in the next version.
>
> ---
>
> **Q5.** "The phrase “efficiency or performance …” in the abstract is unclear."
>
> **A5.** We thank the reviewer for the comment. We will revise the phrase to improve clarity in the updated manuscript.
>
> ---
>
> **In summary,** we have added the comparison with SDTT [1] and other additional experiments, and will include citations and discussions of related works, including [1, 2, 3, 4], in the revised manuscript following your suggestion. We sincerely appreciate the reviewer's helpful feedback. If our responses have addressed your concerns, we would be grateful if you would consider raising the score. Thank you!
>
> **References:**
>
> [1] Deschenaux, Justin, and Caglar Gulcehre. Beyond autoregression: Fast LLMs via self-distillation through time. ICLR 2025.
>
> [2] Sahoo et al. The diffusion duality. ICML 2025.
>
> [3] Sahoo et al. Simple and effective masked diffusion language models. NeurIPS 2024.
>
> [4] Ou et al. Your absorbing discrete diffusion secretly models the conditional distributions of clean data. ICLR 2025.

---

> > ### Author Rebuttal · Reviewer_9z8T · 2026-04-02
> >
> > Thanks for running the requested experiments. Could the authors clarify the experimental details of the SDTT baseline? The (1) training steps (2) Learning rate (3) Batch size (4) Loss function.
> >
> > Was SDTT-LLaDA trained for the same number of steps as d3LLM-LLADA?1
> >
> > Seems like the authors used the reverse KL as the loss function. Can they please report the numbers with the forward KL?
> >
> > I'm quite surprised to see that SDTT-LLada has a worse performance than d3LLM-llada.
> >
> > If the authors address my concerns, im happy to change my score accordingly.

---

> > > ### Author Response · Authors · 2026-04-04
> > >
> > > Thank you for your follow-up questions. We provide detailed clarifications below.
> > >
> > > ---
> > >
> > > ## Fair Comparison:
> > >
> > > We clarify the experimental details of the SDTT baseline as follows:
> > >
> > > - **Training Steps and Batch Size:** To ensure a fair comparison, SDTT-LLaDA and d3LLM-LLaDA were trained under **identical configurations**. Specifically, both methods use the same dataset of 92,428 samples, trained for 6 epochs with a per-device batch size of 4, gradient accumulation steps of 4, and 4 GPUs, resulting in an effective batch size of 64 and a total of 8664 training steps. Both methods employ the AdamW optimizer.
> > >
> > > - **Learning Rate:** We performed a grid search over learning rates {5e-6, 2e-5, 4e-5, 6e-5} for SDTT-LLaDA, where 2e-5 corresponds to d3LLM-LLaDA's setting and 6e-5 follows the recommendation in the original SDTT paper [1]. We report results using the best-performing learning rate, i.e., 2e-5, ensuring that SDTT-LLaDA is evaluated under its optimal configuration.
> > >
> > > The above configurations confirm that the comparison between SDTT and d3LLM is conducted **under fair conditions.**
> > >
> > > ---
> > >
> > > ## Results with Forward KL Loss:
> > >
> > > We appreciate the reviewer's suggestion to evaluate SDTT with forward KL. The original SDTT paper [1] **only used reverse KL across all their experiments**; therefore, we followed their setting and used reverse KL as the loss function in our previous rebuttal.
> > >
> > > Following your suggestion, we have additionally conducted experiments evaluating SDTT with forward KL. The results are shown below (Acc / TPF / AUP):
> > >
> > >   ||LLaDA| SDTT-LLaDA (Bwd KL)|SDTT-LLaDA (Fwd KL)|**d3LLM-LLaDA**|
> > >   |-|:-:|:-:|:-:|:-:|
> > >   |GSM8K-CoT|72.55 / 1.00 / 72.55|71.62 / 4.53 / 309.71|72.17 / 4.29 / 310.33|**73.09 / 9.11 / 637.65**|
> > >
> > > As shown above, SDTT-LLaDA with forward KL achieves slightly higher accuracy than reverse KL. To the best of our understanding, this is likely because forward KL is a zero-avoiding divergence: it penalizes the student when the student assigns low probability to the teacher's high-probability tokens, thereby enforcing a closer approximation of the teacher's distribution at the cost of little parallelism. Nevertheless, d3LLM outperforms both SDTT variants, demonstrating a better trade-off between parallelism and efficiency.
> > >
> > > ---
> > >
> > > ## Why d3LLM Outperforms SDTT:
> > >
> > > We would like to take this opportunity to further discuss the performance advantage of d3LLM.
> > >
> > > - **Effectiveness of Distillation:** First, the above experiments confirm that distillation is effective for accelerating dLLMs in general. For example, SDTT achieves significant speedups with only marginal accuracy degradation over the original LLaDA, validating the value of the distillation paradigm and the effectiveness of SDTT.
> > >
> > > - **Key Advantages of d3LLM:** d3LLM introduces two key improvements in using distillation: (i) *pseudo-trajectory distillation*, which leverages the teacher's decoding order to teach the model which tokens can be decoded confidently at early steps, and (ii) *curriculum learning strategy*, which progressively increases the difficulty of training targets. The ablation studies in our Table 5 demonstrate the effectiveness of each component. Together, these two strategies provide a more refined guidance on distilling the teacher model, thereby achieving better performances.
> > >
> > > To help the reviewer better understand our contribution, we emphasize that our key argument is that **dLLMs should jointly optimize the accuracy–parallelism**. This motivates our proposal of the **AUP metric** with the proposed d3LLM framework to push this trade-off frontier. In this context, we believe that adding a comparison to SDTT would further highlight d3LLM’s advantages. By leveraging the teacher’s decoding order and a curriculum strategy, d3LLM improves distillation performance and, consequently, achieves a better **accuracy–efficiency** trade-off.
> > >
> > > ---
> > >
> > > **In summary**, we will incorporate the implementation details for SDTT-LLaDA and the above additional experimental results into the revised manuscript. We hope these clarifications and results adequately address the reviewer's remaining concerns, and we would be grateful if the reviewer would consider updating the evaluation of our contribution. Thank you!
> > >
> > > **References:**
> > >
> > > [1] Deschenaux, Justin, and Caglar Gulcehre. Beyond autoregression: Fast LLMs via self-distillation through time. ICLR 2025.

---

### Decision · Program_Chairs · 2026-04-30

**Decision:**

Accept (regular)

**Comment:**

This paper addresses an important problem for diffusion LLMs: improving decoding parallelism without sacrificing too much accuracy. The proposed d3LLM framework is well motivated, and the empirical results are strong, showing substantial speedups with competitive performance across multiple models and tasks. Reviewers’ main concerns were about missing baselines, the robustness of AUP, and implementation/efficiency details. In rebuttal, the authors addressed these points well by adding SDTT comparisons, AUP sensitivity analyses, stronger efficiency analysis, and clarifications on KV-refresh and training details. While some components are individually incremental and deeper analysis of pseudo-trajectory distillation would strengthen the paper, the overall contribution is solid and useful to the community. I therefore recommend accept.